# Massive Values in Self-Attention Modules are the Key to Contextual Knowledge Understanding

**Mingyu Jin**[1] **Kai Mei**[1] **Wujiang Xu**[1] **Mingjie Sun**[2]
**Ruixiang Tang**[1] **Mengnan Du**[3] **Zirui Liu**[4] [†] **Yongfeng Zhang**[1] [†]

## Abstract

Large language models (LLMs) have achieved remarkable success in contextual knowledge understanding. In this paper, we show for the first time that these concentrated massive values consistently emerge in specific regions of attention queries (Q) and keys (K) while not having such patterns in values (V) in various modern transformer-based LLMs. Through extensive experiments, we further demonstrate that these massive values play a critical role in interpreting contextual knowledge (i.e., knowledge obtained from the current context window) rather than in retrieving parametric knowledge stored within the model's parameters. Our further investigation of quantization strategies reveals that ignoring these massive values leads to a pronounced drop in performance on tasks requiring rich contextual understanding, aligning with our analysis. Finally, we trace the emergence of concentrated massive values and find that such concentration is caused by Rotary Positional Encoding (RoPE) and it appears since very first layers. These findings shed new light on how Q and K operate in LLMs and offer practical insights for model design and optimization. The code is available at https://github.com/MingyuJ666/Rope_with_LLM.

## 1. Introduction

Large Language Models (LLMs) have achieved remarkable success across various applications (Wei et al., 2022; Brown et al., 2020; Achiam et al., 2023; Sun et al.; Jin et al.; Tang

---
[*] The first two authors are core contributors to this project, listed in random order [†] Correspondence Author [1]Rutgers University [2]Carnegie Mellon University [3]New Jersey Institute of Technology [4]University of Minnesota. Correspondence to: Mingyu Jin <mingyu.jin@rutgers.edu>, Zirui Liu <zrliu@umn.edu>, Yongfeng Zhang <yongfeng.zhang@rutgers.edu>.

*Proceedings of the 42nd International Conference on Machine Learning*, Vancouver, Canada. PMLR 267, 2025. Copyright 2025 by the author(s).

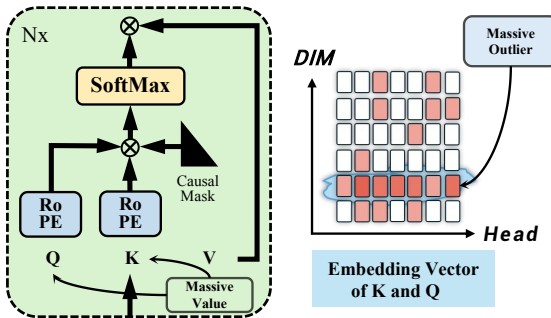

Figure 1. In transformer-based Large Language Models with RoPE (like Llama, Gemma), the attention queries (Q) and keys (K) exhibit concentrated **massive values** in certain dimensions.

et al., 2025; Wang et al., 2025). However, our understanding of their internal mechanisms and how these mechanisms relate to observable behaviors remains limited, posing a challenge to improving their reliability and performance. Researchers have explored LLM representations (Azaria & Mitchell, 2023; Bürger et al., 2024; Gurnee & Tegmark, 2024; Jin et al., 2025) and discovered that residual stream activations can exhibit massive values (i.e., magnitudes significantly larger than typical values (Dettmers et al., 2022; Wei et al., 2021; 2023; Sun et al., 2024; Heo et al., 2024)). Recent studies further (Liu et al.; Hooper et al., 2024; Zhao et al., 2024) observe that massive values appear exclusively in Q and K while are absent in V. Here, by Q, K, and V, we mean the representations output by the query, key, and value layers in self-attention, respectively. These massive values have been identified as critical factors influencing quantization (Liu et al.; Lin et al., 2024), leading to the development of methods such as suppression techniques (Wei et al., 2021; 2023) and protection mechanisms (Lin et al., 2024; Xiao et al., 2023). Although these studies utilize observations of massive values for quantization, they do not explore the rationale behind this counterintuitive phenomenon deeply.

To address this gap, we systematically investigate the formation of massive values and their connection to model behaviors. Our key findings are as follows: ❶ *Massive values are concentrated in specific regions of Q and K exclusively.* This echoes the finding in "Round and Round we go" (Barbero et al., 2025). We observe that massive values concentrate in specific regions of Q and K compu-

tations; *these massive values in each head's dim index are very close* (means concentrate as Definition 2), as is shown in Figure 2. While self-attention computation dictates that each attention head operates independently, the massive values across different heads consistently cluster at remarkably similar positional indices, which is very counterintuitive. This phenomenon is absent in V computations and absent in models without RoPE, such as GPT-2 (Black et al., 2022) and OPT (Zhang et al., 2022). ❷ *Massive values in Q and K are critical for understanding contextual knowledge over parametric knowledge.* Our analysis shows that massive values significantly contribute to contextual knowledge (i.e., knowledge augmented through the context window) rather than parametric knowledge (i.e., knowledge encoded during training). Disrupting these values leads to a notable degradation in tasks requiring contextual understanding, such as passkey retrieval (Jin et al., 2024a; Mohtashami & Jaggi, 2023), IMDB review analysis (Maas et al., 2011), and mathematical reasoning (Cobbe et al., 2021; Ling et al., 2017). In contrast, tasks involving parametric knowledge, such as World-Cities (Marks & Tegmark, 2023), are only subtly affected. Perplexity analysis reveals that disrupting massive values causes a significant decline in reasoning benchmarks, particularly in IMDB tasks, compared to when non-massive values are disrupted. ❸ *Quantization techniques targeting massive values preserve contextual knowledge better.* Our experiments on various quantization methods show that techniques like AWQ (Lin et al., 2024) and SmoothQuant (Xiao et al., 2023), which scale per-channel weights or activations to reduce the dominance of massive values, effectively maintain LLM's contextual knowledge understanding capabilities. In contrast, methods that do not specifically address massive values (Frantar et al., 2022) exhibit significant degradation of LLM's performance in contextual knowledge understanding tasks, while performance in parametric knowledge retrieval is maintained. These findings align with our analysis, which shows that massive values contribute more to LLM's contextual knowledge understanding capabilities. ❹ *Concentration of massive values is caused by RoPE (Su et al., 2024) and it appears since very early layers in LLMs.* We analyze the root cause of concentrated massive values in the Q and K, and investigate layers where massive values become concentrated. Our analysis demonstrates that this phenomenon originates from the mechanisms of RoPE, and the concentration of massive values becomes apparent since the very first layers. Our temporal and causal analysis deepens the understanding of how and when massive values are concentrated.

In summary, our contributions are as follows:

● We systematically investigate the emergence of massive values in attention modules and identify that their concentrations occur exclusively in Q and K, revealing a distinct structural property of these components.

● We analyze the functional role of these massive values in Q and K and conduct extensive experiments on ten datasets to demonstrate that they play a crucial role in enhancing LLMs' ability to process contextual knowledge, while having subtle impact on parametric knowledge retrieval.

● We evaluate three representative quantization methods and show that those explicitly addressing massive values better preserve LLMs' contextual understanding capabilities. This finding reinforces our analysis and provides insights into designing new quantization strategies.

● We conduct the causal and temporal analysis of massive value concentrations, uncovering that they originate from RoPE mechanisms and emerge as early as the initial layers. This deepens the understanding of how massive values form and propagate through the model.

## 2. Preliminary

### 2.1. Position Encoding

**Position Encoding.** Position encoding in transformers can be categorized into two main approaches: absolute and relative position encodings. Absolute position embeddings assign fixed positional vectors, such as sinusoidal embeddings (Vaswani et al., 2017), learned embeddings used in models like GPT-3 (Brown et al., 2020), or dot-product-based adjustments (Ke et al., 2020). In contrast, relative position encodings focus on the distances between tokens and integrate into attention layers, with examples including logit biases like T5 (Raffel et al., 2020; Dai et al., 2019) and rotary position embeddings like RoPE (Su et al., 2024).

**RoPE.** Here, we introduce the basic concept of RoPE. Let's consider a sequence of tokens represented as $w_1, w_2, \cdots, w_L$, and their corresponding embeddings are denoted as $\mathbf{x}_1, \cdots, \mathbf{x}_L \in \mathbb{R}^{|D|}$, where $|D|$ is the dimension of the embedding. The basic idea of RoPE is to incorporate the positional information into the query $\mathbf{q}$ and the key vectors $\mathbf{k}$, respectively. This integration ensures that their inner product $\mathbf{q}^T\mathbf{k}$ will contain the relative positional embedding information inherently. To achieve this, RoPE employs the following vector transformations:

$$\mathbf{q}_m = f_q(\mathbf{x}_m, m) \in \mathbb{R}^{|L|}, \mathbf{k}_n = f_k(\mathbf{x}_n, n) \in \mathbb{R}^{|L|} \quad (1)$$

where $|L|$ is the hidden dimension of each head. The functions $f_q$ and $f_k$, responsible for injecting positional information, are defined as:

$$f_q(\mathbf{x}_m, m) = W_q\mathbf{x}_m e^{im\theta}, f_k(\mathbf{x}_n, n) = W_k\mathbf{x}_n e^{in\theta} \quad (2)$$

where $\theta_d = b^{-2d/|D|}$, $b = 10,000$, and the projection matrices $W_q, W_k : \mathbb{R}^{|D|} \to \mathbb{R}^{|L|}$. RoPE keeps the real part of the inner product $\mathbf{q}^T\mathbf{k}$, which is $\text{Re}(\mathbf{q}^*\mathbf{k})$. This operation ensures that the dot product of the query and key vectors

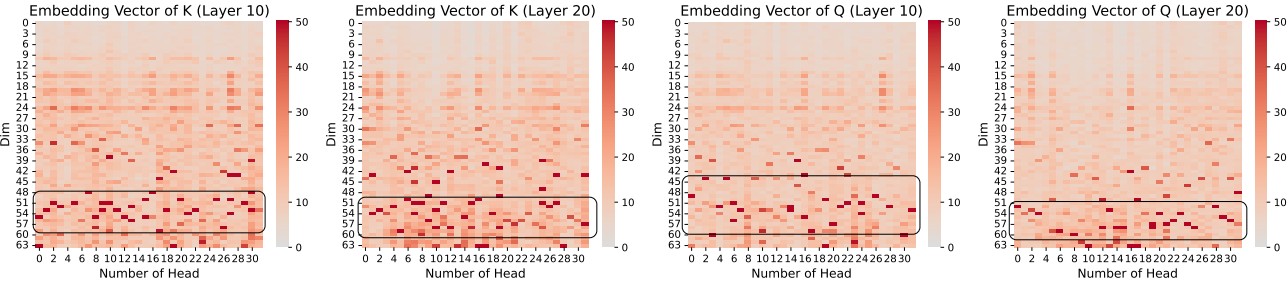

*Figure 2.* Q and K Embedding Vector in Llama-2-7B, we choose Layer 10 and 20, and the input question is shown as Figure 11. This visualization shown here is a two-dimensional image because we averaged over the sequence-length dimension. The horizontal axis is the *number of head* and the vertical axis is *head dim*. We can see that the massive value is concentrated at the bottom of the picture.

depends entirely on the relative distance between the tokens, represented by $m - n$, as follows:

$$\langle f_q(\mathbf{x}_m, m), f_k(\mathbf{x}_n, n) \rangle_{\mathbb{R}} = \text{Re}(\langle f_q(\mathbf{x}_m, m), f_k(\mathbf{x}_n, n) \rangle_{\mathbb{C}})$$
$$= \text{Re}(\mathbf{x}_m^* W_q^* W_k \mathbf{x}_n e^{i\theta(m-n)}) = g(\mathbf{x}_m, \mathbf{x}_n, m-n) \quad (3)$$

where $g(\cdot)$ is an abstract mapping function.

### 2.2. LLM Inference Workflow

LLM inference consists of a prefilling phase that processes the input context and a decode phase that generates tokens autoregressively, with the prefilling phase establishing the initial context representation and the decode phase iteratively producing new tokens based on this foundation.

**Prefilling Phase.** Given the $X \in \mathbb{R}^{l_{\text{prompt}} \times h \times d}$ be the input tensor (we set batch size as 1 by default), $l_{\text{prompt}}$ is the length of the input prompt, $h$ is the number of head, and $d$ is the model hidden size. For convenience, we ignore the layer index here. The key value tensors can be computed by

$$X_K = XW_K, \quad X_V = XW_V, \quad (4)$$

where $W_K, W_V \in \mathbb{R}^{d \times d}$ are the key and value's layer weights, respectively. After obtaining $X_K$ and $X_V$, they are cached in the memory for easy decoding.

**Decoding Phase.** During the decoding phase, let $t \in \mathbb{R}^{b \times 1 \times d}$ be the current input token embedding. Let $t_K = tW_K$ and $t_V = tW_V$ be the key and value layer outputs, respectively. The KV cache is updated by:

$$X_K \leftarrow \text{Concat}(X_K, t_K), \quad X_V \leftarrow \text{Concat}(X_V, t_V), \quad (5)$$

then attention output is calculated as:

$$t_Q = tW_Q, \quad A = \text{Softmax}(t_Q X_K^\top), \quad t_O = AX_V \quad (6)$$

where $W_Q$ is the weight matrix of the query layer. For ease of illustration, we ignore the attention output layer and other parts of the inference workflow.

## 3. The Effect of Massive Values in Knowledge Understanding

### 3.1. Massive Value

The attention queries (Q) and keys (K) in mainstream LLMs are typically represented as $Q, K \in \mathbb{R}^{\mathcal{B} \times \mathcal{S} \times \mathcal{H} \times \mathcal{D}}$, where $\mathcal{B}$ is the batch size, $\mathcal{S}$ is the sequence length, $\mathcal{H}$ is the number of attention heads, and $\mathcal{D}$ is the head dimension. Assuming $\mathcal{B} = 1$, we compute the L2 norm along the sequence length dimension, reducing the representation to a matrix $M \in \mathbb{R}^{\mathcal{H} \times \mathcal{D}}$, where each element $M_{h,d}$ represents the norm of the corresponding component in head $h$ and dimension $d$. Taking $Q$ as an example, $M_{h,d}$ is computed as:

$$M_{h,d} = \|Q_{:,h,d}\|_2 = \sqrt{\sum_{s=1}^{\mathcal{S}} Q_{s,h,d}^2}. \quad (7)$$

**Definition 1.** *(Massive Value) A massive value is an element $M_{h,d}$ that satisfies:*

$$M_{h,d} > \lambda \frac{1}{\mathcal{D}} \sum_{d'=1}^{\mathcal{D}} M_{h,d'} \quad (8)$$

*where $\lambda > 1$ is a threshold controlling massive value selection. In our experiments, we empirically set $\lambda = 5$.*

**Definition 2.** *(Concentrate Massive Value) In each attention head, certain dimensions exhibit notably massive values, and these tend to cluster in specific dimensional regions (shown in the black box in the Figure 2). Moreover, across different attention heads, these massive values often appear in similar positions.*

### 3.2. Contextual Knowledge Understanding and Parametric Knowledge Retrieval

Research demonstrates that Large Language Models (LLMs) acquire extensive knowledge through pre-training on large-scale corpora, termed Parametric Knowledge (PK) (Cheng et al., 2024; Xu et al., 2024). In practical applications, LLMs augment this parametric knowledge (PK) with Contextual

Knowledge (CK)—additional information provided within the input context (Xu et al., 2024; Antoniades et al., 2025). This distinction gives rise to two fundamental tasks:

**Remark 3.1.** *(Contextual Knowledge Understanding) refers to understanding the content within a paragraph and using the information it provides to answer questions. For example, it could involve identifying a key amidst a collection of meaningless text or determining whether a movie review is positive or negative based on the content of the review (Xu et al., 2024; Cheng et al., 2024; Wen et al., 2024).*

**Remark 3.2.** *(Parametric Knowledge Retrieval) refers to questions that can be answered correctly by simply using the query and the knowledge within the model to perform a retrieval match. For example, "What is the capital of the United States?" The answer would be "Washington D.C." (Xu et al., 2024; Cheng et al., 2024)*

**Dataset:** We separate datasets into the two main categories corresponding to Remark 3.1 and Remark 3.2. For ***Contextual Knowledge Understanding Tasks***, we adopt mathematical reasoning benchmarks (i.e., GSM-8K (Cobbe et al., 2021), AQUA (Ling et al., 2017)), sentiment analysis dataset (i.e., IMDB (Maas et al., 2011)), and synthetic passkey retrieval datasets as Appendix A in different difficulty levels. For ***Parametric Knowledge Retrieval Tasks***, we adopt factual knowledge QA such as Cities (Marks & Tegmark, 2023) and our synthetic datasets covering topics in Sports, Arts, Technology, and Celebrity. The rationale behind our synthesis choice is that we would like to use simpler and direct queries that focus on direct factual knowledge rather than being blended with reasoning. The dataset details and our data synthesis pipeline can be found at Appendix E and Appendix A.

### 3.3. Disruption of Massive Value

To investigate the impact of massive values in *Definition 1* on LLM performance, we systematically examine the effects of disrupting both massive values and non-massive values, respectively, in LLMs equipped with RoPE. The disruption is carefully synchronized with the LLM's two-phase like Section 2.2 content generation process to ensure accurate manipulation of massive values at the appropriate stage.

**Disruption Setup.** Our disruption of massive values and non-massive values is specifically targeted at the **prefilling stage**, as disrupting both prefilling and decoding stages would compromise the model's fundamental language modeling capabilities. This selective approach ensures that any observed performance changes can be attributed to the impact on context processing rather than impairment of the model's generative abilities. For example, let $\mathbf{X} \in \mathbb{R}^{l \times h \times d}$ denote the query tensor. We replace the values at the massive value indices with the average value computed over the query tensor like Equation 9.

$$\mathbf{X}_{i;j;k^*} = \begin{cases} \text{Mean}(\mathbf{X}) \,, & k^* = \text{argmax}_k x_{i;j;k} \\ \mathbf{X}_{i;j;k^*} \,, & k^* \neq \text{argmax}_k x_{i;j;k} \end{cases} \quad (9)$$

Our investigation reveals that disrupting massive values can be accomplished through several substitution methods: using mean values, zeros, maxima, or minima. We disrupt massive/non-massive values *on both Q and K* like Table 1. More comprehensive experiments comparing these replacement strategies are detailed in Section 3.4.

### 3.4. Massive Values Contribute to Contextual Knowledge Understanding

As shown in Table 1, *Parametric Knowledge Retrieval* tasks still maintain *relatively high accuracy* even when massive values are disrupted, demonstrating performance comparable to their original unmodified state and when non-massive values are disrupted. Cities tasks maintain relatively strong performance with scores achieving from 76% to 88%, showing a degradation of only 15-20%. Sports and Art categories show moderate impact, with performance dropping to the 65-75% range. Technology maintains similar resilience levels, with scores remaining above 70%. The Celebrity category shows the strongest resilience, maintaining performance above 70% across all models. Importantly, when *only non-massive values* are disrupted, performance remains remarkably stable across all tasks and models, with variations typically less than $\pm 1\%$. This striking contrast suggests that *massive values play a crucial role specifically in **contextual knowledge understanding** tasks, while **parametric knowledge retrieval** tasks can maintain reasonable performance even when these values are disrupted.* For mathematical reasoning tasks, the impact is devastating: GSM8K accuracy drops dramatically (Gemma2-9B: 81.30% to 15.10%, Llama3-8B: 76.90% to 4.00%, Qwen2.5-7B: 86.60% to 16.10%), and AQUA shows similar degradation with performance declining by over 40 percentage points. Passkey Retrieval tasks collapse from 100% to near-zero accuracy (0-2%) across all difficulty levels, while IMDB sentiment analysis performance plummets from above 94% to single digits across all models.

**PPL and Diversity Score.** In addition to accuracy, we evaluate perplexity (PPL) and diversity as complementary metrics to understand the impact of massive values on model performance. *These metrics reinforce our previous findings regarding the crucial role of massive values in contextual knowledge understanding tasks.* Perplexity, which measures a language model's prediction confidence (lower values indicating better language modeling ability), and diversity, quantified through 2-gram diversity scores to assess the richness and variety of model outputs, both demonstrate patterns consistent with our accuracy-based observations. We analyzed the model's *perplexity (PPL) and diversity scores*

*Table 1.* Results of LLMs under different settings (vanilla, massive value disrupted, non-massive value disrupted) on different benchmarks. For the Passkey Retrieval Task, the values (max prompt token length, passkey length) represent the maximum number of tokens allowed in the prompt and the length of the passkey to be retrieved, respectively. All values are reported in percentage (%).

| Model | Contextual Knowledge Understanding Task | | | | | | Parametric Knowledge Retrieval Task | | | | |
| --- | --- | --- | --- | --- | --- | --- | --- | --- | --- | --- | --- |
| | GSM8K | AQUA | Passkey Retrieval Task | | | IMDB | Cities | Sports | Art | Technology | Celebrity |
| | | | (128,6) | (256,12) | (1024,48) | | | | | | |
| *Gemma2-9B* | 81.30 | 63.80 | 100 | 100 | 100 | 94.70 | 99.70 | 91.00 | 84.00 | 81.00 | 92.50 |
| + Non-Massive Value Disrupted | 81.60 | 65.60 | 100 | 100 | 100 | 97.40 | 99.60 | 91.00 | 84.00 | 81.50 | 92.50 |
| + Massive Value Disrupted | 15.10 | 16.50 | 2.00 | 0.00 | 0.00 | 1.80 | 76.40 | 73.50 | 68.00 | 72.00 | 82.00 |
| *Llama3-8B* | 76.90 | 53.51 | 100 | 100 | 100 | 95.40 | 99.40 | 95.00 | 93.50 | 92.50 | 95.00 |
| + Non-Massive Value Disrupted | 77.40 | 53.90 | 100 | 100 | 100 | 95.40 | 99.40 | 94.50 | 93.00 | 92.50 | 95.50 |
| + Massive Value Disrupted | 4.00 | 9.68 | 9.00 | 0.00 | 0.00 | 11.00 | 88.20 | 74.50 | 64.00 | 74.90 | 73.00 |
| *Qwen2.5-7B* | 86.60 | 56.69 | 100 | 100 | 100 | 96.80 | 97.70 | 95.00 | 96.00 | 90.00 | 93.50 |
| + Non-Massive Value Disrupted | 85.40 | 57.28 | 100 | 100 | 100 | 97.60 | 97.50 | 94.00 | 96.50 | 90.00 | 93.50 |
| + Massive Value Disrupted | 16.10 | 19.68 | 9.00 | 1.00 | 0.00 | 6.53 | 81.50 | 74.00 | 69.50 | 71.00 | 71.00 |

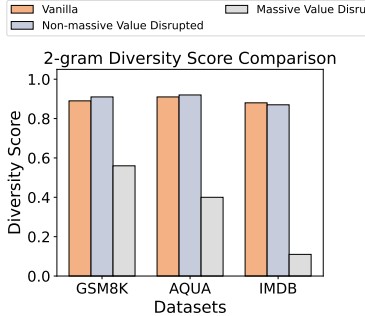

(a) The 2-gram diversity scores for Llama3-8B under three conditions across three datasets.

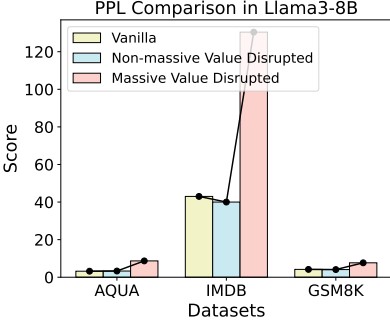

(b) The perplexity scores of the Llama3-8B model under three conditions across three datasets.

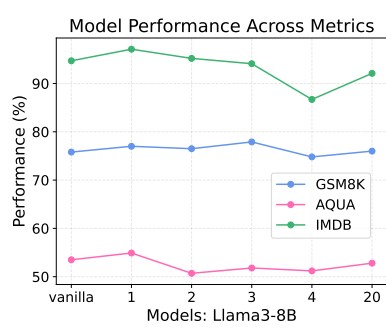

(c) The figure shows Llama3-8B maintains relatively performance under different non-massive value disrupted methods.

*Figure 3.* Disrupting massive values leads to higher perplexity and lower diversity, while disrupting non-massive values maintains model performance, particularly evident in IMDB dataset analysis.

before and after disrupting massive values, with particular focus on Llama3-8B as shown in Figure 3b and Figure 3a (additional results can be found in Table 7). The experimental results reveal striking patterns: When massive values are disrupted, the LLM shows severely degraded performance. Particularly in the IMDB dataset, we observe a dramatic increase in perplexity (∼80) and a decrease in 2-gram diversity scores (∼0.1) when massive values are disrupted, while vanilla and non-massive value disrupted conditions maintain low perplexity and high diversity (∼0.9). *These findings further confirm that massive values are crucial for contextual knowledge understanding tasks.*

**Controlled Experiment.** We conduct our disruption of massive values and non-massive values specifically during the **prefilling stage**. For non-massive values disruption, like the process in massive value disruption, we let $\mathbf{X} \in \mathbb{R}^{l \times h \times d}$ represent the query tensor, and we replace the smallest values with calculated averages. Specifically, we identify the top n smallest values along the last dimension ($k$) and replace them with the mean value computed across the entire query tensor. For presentation clarity, we omit the batch dimension as these operations are applied consistently.

$$\mathbf{X}_{i;j;k^*} = \begin{cases} \text{Mean}(\mathbf{X}), \ k^* = \arg\min x_{i;j;k} \\ \mathbf{X}_{i;j;k^*}, \ k^* \neq \arg\min x_{i;j;k} \end{cases} \quad (10)$$

As shown in Figure 3c, we systematically varied n from 1 to 20 (represented on the horizontal axis) in our control experiments. The results demonstrate that *replacing these minimum values with averages has a subtle impact on model performance across all three contextual knowledge understanding benchmarks*, with GSM8K, AQUA, and IMDB maintaining relatively stable performance throughout the range of n values tested (additional results in Appendix F). This robustness to non-massive value manipulation provides compelling evidence that *the model's contextual knowledge understanding capabilities **primarily rely on regions that massive values concentrate***.

**Disruption Methods Show Consistent Effects.** Different approaches to disrupting massive values (using mean, zero, or minimum value substitution like mentioned in Section 3.3) yield consistently similar effects across benchmarks. For instance, in GSM8k, all disruption methods reduce performance to single digits (mean: 4.00%, zero: 1.60%,

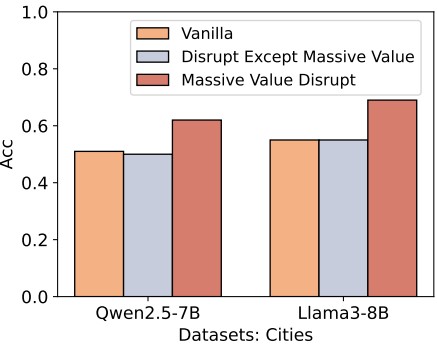

*Figure 4.* We can observe that introducing conflicting background knowledge causes LLM to be misled into making random guesses. However, **after massive values are disrupted, the model is still able to maintain a certain level of accuracy.**

minimum: 2.73%), while non-massive value manipulations maintain performance close to the vanilla baseline (76-77%). This pattern holds across all benchmarks, demonstrating that the *impact of massive value disruption is robust to the variations of the disruption method.* Additional results across other models can be found in Appendix F.

*Table 2.* Impacts of applying different methods to disrupt massive values/non-massive values on model performance (Llama3-8b), where ✓ indicates disrupting massive value, while ✗ indicates disrupting non-massive value.

| Disruption | Disruption Condition | GSM8k | AQUA | IMDB | Passkey Retrieval | |
|---|---|---|---|---|---|---|
| | | | | | (128,6) | (256,12) |
| None | – | 76.90 | 53.51 | 95.40 | 100 | 100 |
| Mean | ✓ | 4.00 | 9.68 | 11.00 | 9.00 | 0.00 |
| | ✗ | 77.40 | 53.90 | 95.40 | 100 | 100 |
| Zero | ✓ | 1.60 | 8.07 | 13.40 | 8.00 | 0.00 |
| | ✗ | 76.40 | 53.83 | 94.70 | 100 | 100 |
| Min | ✓ | 2.73 | 9.33 | 10.80 | 9.00 | 0.00 |
| | ✗ | 75.20 | 53.98 | 95.40 | 100 | 100 |

### 3.5. Effects of Massive Values on Knowledge Conflict.

Our discovery of the differential impact of extreme values on contextual knowledge understanding versus parametric knowledge retrieval tasks motivated a novel experimental investigation. We designed an experiment to *introduce conflicting contextual information* in factual datasets, particularly focusing on geographical knowledge from cities (Marks & Tegmark, 2023; Longpre et al., 2021). For instance, we modified straightforward questions like "Is New York a city in the United States?" by adding contradictory context: "Geographical knowledge has changed, and New York has become a city in the United Kingdom. Is New York still a city in the United States?"

As shown in Figure 4, we tested this modified cities dataset across three model conditions. Interestingly, while the vanilla models (Llama3-8B and Qwen2.5-7B) achieve approximately 50% accuracy on this binary classification task - equivalent to random guessing - the models with massive

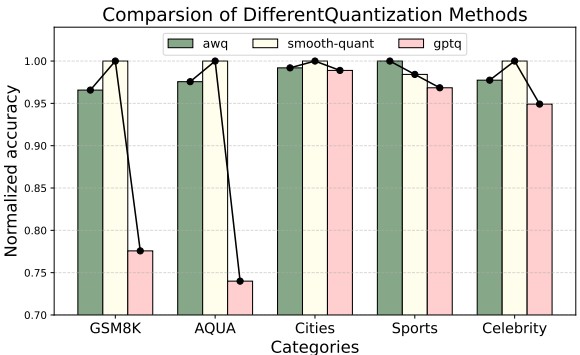

*Figure 5.* Impacts of different quantization methods on Llama3-8b across different benchmarks.

values disrupted demonstrate accuracy significantly above chance level. This *counter-intuitive improvement suggests that when massive values are destroyed, the model loses its ability to process misleading contextual information and instead defaults to its parametric knowledge*, effectively ignoring the contradictory context. This observation aligns with previous findings (Yu et al.) that suggest distinct regions within neural networks are responsible for context understanding versus parametric knowledge storage.

### 3.6. The Effects of Massive Values on Quantization

To further validate our findings regarding the importance of massive value in model behavior, we evaluated three well-established quantization methods: AWQ (Lin et al., 2024), SmoothQuant (Xiao et al., 2023), and GPTQ (Frantar et al., 2022). As shown in Figure 5, these methods demonstrate distinctly different patterns across task types.

AWQ and SmoothQuant, which explicitly *preserve massive values* during quantization, maintain strong performance across all tasks. AWQ achieves this by selectively protecting "important" weights during quantization, while SmoothQuant employs a smoothing factor ($S$) to redistribute massive values in activations through mathematically equivalent transformations. In contrast, GPTQ, which does *not specifically protect massive values*, shows significant performance *degradation on contextual knowledge understanding tasks*, particularly on GSM8K and AQUA (dropping to approximately 75% normalized accuracy), while maintaining comparable performance on parametric knowledge retrieval tasks (Cities, Sports, Celebrity). This performance disparity is particularly revealing: quantization methods that protect massive values maintain good and robust performance on contextual knowledge understanding tasks, while methods that do not protect these values struggle specifically with contextual knowledge understanding tasks. *These results provide additional evidence for our hypothesis about the **crucial role of massive values in LLM's contextual knowledge understanding capabilities.***

# 4. Causal Mechanisms and Temporal Analysis of Concentrated Massive Values

## 4.1. RoPE Contribute to Concentrated Massive Values

Building upon existing research documenting the relationship between RoPE and the embedding patterns of concentrated massive values (in *Definition* 1 and 2) (Barbero et al., 2025), we continue investigating this phenomenon in $Q$ and $K$ embeddings. Based on several observations and analyses, we propose three hypotheses suggesting that the concentration massive values is an inherent consequence of RoPE's implementation.

**Evidence 1:** In large language models (LLMs), the Rotary Position Embedding (RoPE) implements a sophisticated dimensional pairing approach to encode positional information. The mechanism divides the embedding dimensionality ($d$) into ($\lceil \frac{d}{2} \rceil$) pairs, where each pair undergoes rotation at a unique frequency determined by its dimensional index.

*Frequency Distribution:* For each dimensional pair ($j$), the rotation frequency is governed by the equation:

$$\theta_j = 10000^{-2j/d} \tag{11}$$

This formulation creates an exponential decay in frequencies across dimensional pairs, *resulting in a spectrum from high-frequency to low-frequency rotations.*

*Dimensional Rotation:* Each pair of dimensions ( ($x_{2j}$, $x_{2j+1}$) ) undergoes rotation according to the matrix operation:

$$\begin{bmatrix} \cos(m\theta_j) & -\sin(m\theta_j) \\ \sin(m\theta_j) & \cos(m\theta_j) \end{bmatrix} \begin{bmatrix} x_{2j} \\ x_{2j+1} \end{bmatrix} \tag{12}$$

where $m$ represents the position in the sequence.

*Multi-Scale Position Sensitivity:* This frequency distribution serves two crucial purposes: *High-frequency components (small $j$) enable fine-grained position discrimination at local scales*, as the rotation angle changes rapidly with position. Low-frequency components (large $j$) capture long-range positional relationships as their rotation angles change gradually across positions. The resulting position-encoded representations interact with the attention mechanism to produce position-aware similarity scores. It decreases from 0-($\lceil \frac{d}{2} \rceil$) as the dimensional rotation angle decreases, causing the massive values to concentrate at the bottom (i.e., low-frequency regions) due to less position information possessed. This concentration of massive values in low-frequency regions primarily encodes rich semantic content rather than positional information, as evidenced by experimental results showing that **disrupting these values severely impairs contextual understanding tasks** (with IMDB accuracy dropping from 94% to single digits) **while**

**preserving basic parametric knowledge retrieval** (showing only 15-20% degradation). The dramatic increase in perplexity and decrease in output diversity when massive values are disrupted further confirms their crucial role in semantic comprehension rather than positional encoding.

**Evidence 2:** RoPE applies position encoding selectively: it operates exclusively on $K$ and $Q$, but not on $V$. As explained in section 2, this design is intentional—RoPE specifically incorporates positional information into the query **q** and key vectors **k**, enabling their inner product $\mathbf{q}^T\mathbf{k}$ to inherently capture relative positional relationships. This selective application has a direct consequence: ***the concentrated massive value appear exclusively in the $Q$ and $K$, while being completely absent in $V$***, as demonstrated in Figure 7. This pattern is consistently observed across various architectures—as shown in Table 3, all LLMs implementing RoPE exhibit these concentrated massive values in the $Q$ and $K$, while $V$ shows no discernible regular patterns.

**Evidence 3:** *Concentrated Massive Value in Q and K appears exclusively in the LLM with RoPE like Gemma*: To validate our hypothesis about the relationship between RoPE and massive values, we conducted a comparative analysis across different model architectures. First, we examined models without RoPE, including OPT (Zhang et al., 2022) and GPT-2 (Black et al., 2021). As expected, their embedding vector maps showed no signs of organized massive values, with OPT-2.7B's embedding vectors (as Figure 6) appearing particularly disorganized and chaotic. In contrast, we observed consistent patterns of massive values in models employing RoPE, extending beyond traditional language models (like Llama (Dubey et al., 2024), Qwen (Yang et al., 2024)) to multimodal architectures. For instance, both LLaVA-1.5 (Liu et al., 2024) and Qwen2-VL (Wang et al., 2024) exhibit similar massive value patterns (as shown in Figure 22 in the Appendix). Notably, Qwen2-VL implements Multimodal Rotary Position Embedding (M-RoPE), which decomposes positional embedding to capture 1D, 2D, and 3D positional information yet maintains the characteristic massive value patterns. Under controlled conditions, *GPT2-NEO and GPT2-NEOX share a similar architecture, with the only difference being the presence of RoPE in GPT2-NEOX. While GPT2-NEO does not exhibit concentrated extreme values, GPT2-NEOX clearly does.* But the RoPE in GPT2-NEOX is not similar to the RoPE applied in Llama and Gemma, the concentrated massive value will appear in K, Q embedding space, but the position is different; the pattern is shown in Figure 27 in Appendix C and Appendix F.

Further supporting our hypothesis, ***models using alternative position encoding mechanisms***, such as Jamba (Lieber et al., 2024), which combines Mamba and Transformer architectures without RoPE, ***show no evidence of these massive value patterns.*** Detailed visualizations and more analysis

| Model | RoPE | Concentrated Massive Values |
|---|---|---|
| Llama 2, 3 | ✓ | ✓ |
| Qwen 2, 2.5 | ✓ | ✓ |
| Gemma 1, 2 | ✓ | ✓ |
| Phi-3 | ✓ | ✓ |
| Falcon3 | ✓ | ✓ |
| LLAVA | ✓ | ✓ |
| Qwen2-VL | ✓ | ✓ |
| Mistral-v0.3 | ✓ | ✓ |
| GPT-NeoX | ✓ | ✓ |
| GPT-2 | ✗ | ✗ |
| GPT-Neo | ✗ | ✗ |
| OPT all size | ✗ | ✗ |
| Jamba | ✗ | ✗ |

*Table 3.* Analysis of different models about whether they adopt RoPE and whether concentrated massive values in Q and K can be observed in these models.

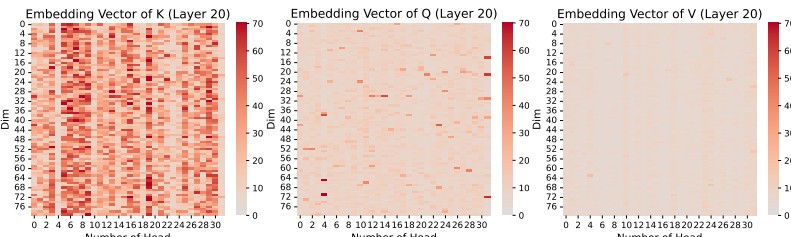

*Figure 6.* K, Q and V in Layer 20 of OPT-350M, where massive values are not concentrated in specific regions.

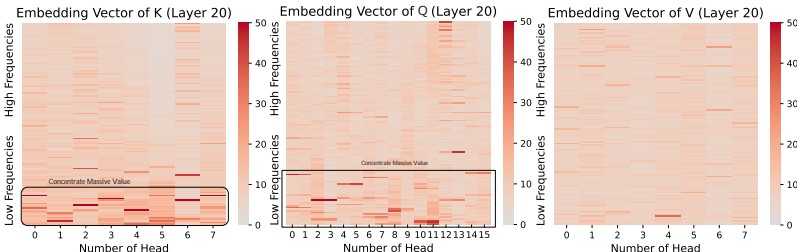

*Figure 7.* K, Q, and V at Layer 20 in Gemma2-9B. Massive values are concentrated in low-frequency regions of K and Q, absent in V.

of LLMs employing different position embeddings that can support these findings are provided in the Appendix C.

**Evidence 4:** Since RoPE divides the dimensionality into pairs and applies similar rotational operations across these pairs, we hypothesized that the pattern of massive values should exhibit a mirrored structure across the two halves of the embedding dimensions. Indeed, our experimental results confirm this prediction: we observe two distinct clusters of massive values in the embedding vector, with one cluster appearing in the first half of the dimensions and a corresponding cluster in the second half, creating a symmetric pattern. This symmetrical distribution of massive values aligns perfectly with RoPE's dimensional pairing mechanism and provides additional evidence that these patterns are not random artifacts but rather emerge from the fundamental architectural design of positional encoding in transformer models.

### 4.2. When Concentrated Massive Values Occur

Through extensive experiments and observations, we found that LLMs such as Llama, Gemma, and so on exhibit massive values in Q and K starting from the very first layer, which is different from massive attentions (Sun et al., 2024) that become significantly apparent only in the later layers. However, there are subtle differences in the patterns between each layer. The layer-wise visual illustrations can be found in Appendix F. In each layer, when extracting the Q and K, we consistently observe that there are no significant differences between the results taken before and after applying RoPE. The patterns remain largely consistent across all layers. This indicates that the massive value is gradually

formed through training rather than being solely caused by the addition of RoPE (since RoPE itself is not trainable).

## 5. Related Work

### 5.1. LLM Quantizations and Massive Values in LLM

Various existing works in quantization have studied the existence of outlier (also called massive value) features in LLMs (Dettmers et al., 2022; Ahmadian et al., 2023; Guo et al., 2024; Xu et al., 2025b). Some research showed that outlier features have large activation values in most of their sequence dimensions. BERT (Devlin et al., 2019) often focuses attention on the "[SEP]" token (Clark et al.; Xiao et al., 2024), while LLMs predominantly assign attention to the starting word token (Xiao et al., 2024); some research focuses on the massive activation value in activation in LLM (Sun et al., 2024) and identified attention artifacts in ViTs (Darcet et al., 2024). Some studies also try to understand the mechanism of these outlier feature in transformer (Guo et al., 2024).

Research on LLM quantization has predominantly focused on analyzing model weights and activations through two main approaches: (1) W8A8 quantization, which converts both activations and weights to INT8 (Wei et al., 2021; 2023; Xiao et al., 2023; Yao et al., 2022; Dettmers et al., 2022), and (2) low-bit weight-only quantization (e.g., W4A16), which converts only weights to low-bit integers (Frantar et al., 2022; Lin et al., 2024; Sheng et al., 2023; Park et al., 2024). During quantization, some studies have identified outlier feature values in activation values (Liu et al.; Sun et al., 2024) as a critical factor that can affect quantization. Several approaches have been developed to handle

outlier feature specifically, either specific suppression techniques (Wei et al., 2021; 2023) or protection mechanisms (Lin et al., 2024; Xiao et al., 2023) to maintain outlier feature unchanged. Quantization methods specifically address outlier feature that are typically analyzed within individual attention heads. Our work examines massive values across the depth dimension of multiple attention heads, offering potential new directions for future quantization approaches.

### 5.2. Rotary Positional Encoding

Rotary Positional Encoding (RoPE), introduced by Su et al. (2024), has become a standard component in major LLMs including Llama (Touvron et al., 2023), Palm (Anil et al., 2023), Mistral (Jiang et al., 2023), and GLM (Du et al., 2022). Studies on long-context Transformers (Sun et al., 2023; Xiong et al., 2024; Dubey et al., 2024) demonstrate that increasing RoPE's $\theta$ parameter from 10,000 to larger values (e.g., 500,000) reduces attention decay and improves learning over extended contexts. Qwen2-VL advances this with M-ROPE, decomposing positional embedding into 1D, 2D, and 3D components for enhanced multimodal processing (Wang et al., 2024). In parallel, research on NoPE (No Positional Encodings) has shown promising results in out-of-distribution (OOD) settings compared to RoPE, suggesting that the causal mechanism alone might sufficiently capture positional information without explicit position encoding (Haviv et al., 2022; Kazemnejad et al., 2024). Analysis of Gemma-7b reveals that embedding vectors' high-frequency components encode position while low-frequency parts carry semantics (Barbero et al., 2025). Some researchers replaced the original position embedding of GPT-2 with rotational position embedding (RoPE), following the standard implementation of GPT-NeoX, which resulted in better training results (Allen-Zhu & Li). Our work explores RoPE's contribution to value formation in transformer-based LLMs.

## 6. Conclusion

Our study provides novel insights into the role and origin of massive values in Large Language Models (LLMs). Through systematic investigation, we find that massive values are critical in contextual knowledge understanding tasks, such as passkey retrieval and IMDB sentiment understanding. In contrast, their influence on parametric knowledge retrieval tasks, such as world knowledge retrieval, is limited. This finding emphasizes the importance of preserving massive value to maintain model performance in reasoning and context-dependent tasks. Our investigation reveals that RoPE induces massive value stripes, distinct patterns exclusively in the Q and K, while absent in models without RoPE, such as OPT. This highlights how positional encoding mechanisms contribute to massive values, particularly low-frequency channel dimensions, offering new insights

into RoPE's role in LLMs. This study establishes a deeper understanding of massive values in LLMs, their critical role in contextual knowledge understanding, their implications for model optimization techniques such as quantization, and their connection to RoPE-induced patterns. These findings lay the foundation for developing more robust, efficient, and interpretable LLM architectures and optimization strategies.

## Acknowledgment

We thank Taowen Wang, Cheng Han, Wenyue Hua, Fei Sun, Hongye Jin, Yixuan Tang and Fan Yang for their valuable discussions and suggestions during the project.

## Impact Statement

This work advances our understanding of Large Language Models by systematically investigating massive values in attention mechanisms and their relationship with contextual knowledge understanding. Our findings provide valuable insights for future model design and optimization, particularly in model quantization and parameter-efficient fine-tuning areas. Our discoveries about the relationship between massive values and contextual knowledge understanding could inform more efficient model compression techniques, potentially reducing the computational resources required to deploy these models. This could help make language models more accessible while maintaining their core capabilities. While this technical advancement in understanding model internals has primarily positive implications for model efficiency and effectiveness, we acknowledge that improvements in LLM capabilities may have broader societal impacts that warrant ongoing discussion and careful consideration by the research community. Our work is focused on advancing the fundamental understanding of machine learning systems, particularly in transformer architectures and attention mechanisms. The ethical implications and societal consequences align with those generally associated with improvements in LLM understanding and optimization.

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

# APPENDIX

This appendix contains additional details for the paper. The appendix is organized as follows:

- Section reports §A more about **Data Synthesis Pipeline**.

- Section reports §B more about **More visualization results for LLMs without RoPE**.

- Section reports §C more about **More visualization results for LLMs with RoPE**.

- Section reports §D more about **Some Results when massive values are disrupted in LLMs.**.

- Section reports §E more about **Experiment Details**.

- Section reports §F more about **More Experiments to Support our Conclusion**.

## A. Data Synthesis Pipeline

### A.1. Synthesizing Parametric Knowledge Retrieval Dataset

The pipeline for synthesizing parametric knowledge retrieval datasets as shown as below. Following existing benchmarks, we choose the commonly-used topics covering Sports, Arts, Technology and Celebrity to feed into the LLM to generate factual knowledge QA related to these topics (Hu et al., 2023; Xu et al., 2025a). For the QA pair, the LLM is required to generate true/false following predefined formatting requirements like (Zheng et al., 2023). These generated questions are verified by another LLM instance regarding each question's structure, answer, and correctness, making improvements when necessary to meet quality requirements (Zheng et al., 2023; Chiang & Lee, 2023). To ensure the highest standards of data quality, a manual inspection stage is incorporated, where human annotators review and validate the generated questions, marking them as either acceptable or requiring revision.

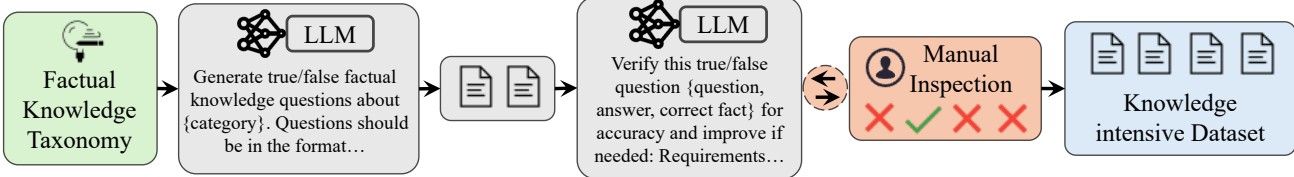

*Figure 8.* **Pipeline of synthesizing parametric knowledge retrieval dataset:** The process starts with a factual knowledge taxonomy as input. An LLM generates true/false factual questions in a predefined format. Another LLM verifies and refines these questions for accuracy. A manual inspection ensures quality, producing a knowledge-intensive dataset for parametric knowledge retrieval tasks.

To ensure the quality of our synthesized question-answer pairs, we implement a two-stage verification pipeline. First, we leverage LLM (i.e., claude-3.5-sonnet) as an initial filter to automatically assess the factuality of the generated content across different categories (Sports, Arts, Technology, and Celebrity). This automated verification serves as an efficient preliminary screening mechanism. Subsequently, we conduct a rigorous manual inspection by human evaluators to further validate the filtered data by the LLM evaluator to avoid hallucinations. After the human evaluation, we sample 200 examples for each category to construct the final synthetic dataset. The verification statistics are shown in Table 4, and some examples of the factual QA are shown in Table 5.

*Table 4.* Verification of whether the synthesized question and answer fulfills the factuality by both LLM and human.

| Category | LLM Verification | | Manual Inspection | |
|---|---|---|---|---|
| | Pass Rate | Failed Case | Pass Rate | Failed Case |
| Sports | 100% | 0% | 92% | 8% |
| Arts | 98% | 2% | 89% | 11% |
| Technology | 99% | 1% | 90% | 10% |
| Celebrity | 99% | 1% | 90% | 10% |

Following the two-stage pipeline, we need to use GPT4 and Human to evaluate and check the dataset, to verify whether our synthetic dataset is completely knowledge-based, without any context understanding (Jin et al., 2024c;b). We observed a little difference in pass rates between the LLM and human inspections. As shown in Table 4, while the LLM achieved near-perfect accuracy with a pass rate of 98-100% across all categories, manual inspection revealed discrepancies, with pass rates ranging from 89% to 92%. This highlights the necessity of human evaluation to catch subtle errors, especially in nuanced categories like Arts and Celebrity, where subjective interpretations and context might introduce inaccuracies. The example in our dataset can be seen at Table 5. These carefully curated question-answer pairs form a robust foundation for downstream tasks such as knowledge-intensive evaluations and parametric knowledge retrieval testing.

*Table 5.* Examples of parameter knowledge retrieval task: factual QA, covering Sports, Arts, Technology and Celebrity.

| Category | Example | Ground Truth |
|---|---|---|
| Sports | Is the Olympic Games held every four years? | Yes |
| | Was Babe Ruth a famous football player? | No |
| | Is the FIFA World Cup held every two years? | No |
| Arts | Was the painting 'Girl with a Pearl Earring' completed during the 18th century? | No |
| | Is Pablo Picasso one of the founding figures of Cubism? | Yes |
| | Was Diego Rivera a famous Mexican muralist? | Yes |
| Technology | Is the ASCII character set limited to 256 characters? | No |
| | Was the first iPhone released in 2007? | Yes |
| | Is Linux an open-source operating system? | Yes |
| Celebrity | Is Leonardo DiCaprio an Oscar-winning actor? | Yes |
| | Was Taylor Swift born in Los Angeles? | No |
| | Was Michael Jackson a member of The Beatles? | No |

### A.2. Synthesizing Passkey Retrieval Dataset

The passkey retrieval task is the same as defined in Landmark Attention (Mohtashami & Jaggi, 2023), a synthetic long context task. It requires a language model to retrieve a simple passkey (i.e., a 6-digit random number) from a long meaningless text sequence. The passkey is placed at various document depths and context lengths (Jin et al., 2024a) (maybe ranging from 4k to 24k). The LLM's performance on this is not sensitive to the prompt (the prompt is all garbled, the LLM is not sensitive to these different prompts) (Anthropic, 2023). This is likely because the sentence carrying the passkey is significantly different from the surrounding repeated random text. Empirically, within the effective context window, almost all LLMs, including those without any instruction tuning or alignment, are able to locate the sentence carrying the passkey.

---

**An example of Passkey Retrieval Task (6-128)**

**Prompt:** There is important info hidden inside a lot of irrelevant text. Find it and memorize it. I will quiz you about the important information there. The grass is green. The sky is blue. The sun is yellow. Here we go. There and back again. The grass is green. The sky is blue. The sun is yellow. Here we go. There and back again. The grass is green. The sky is blue. The sun is yellow. Here we go. There and back again. The pass key is 383816. Remember it. 383816 is the passkey.
**Ground Truth:** 383816

---

*Figure 9.* An example in Passkey Retrieval Task, this is the easier case in this kind of task. The pass key length is 6 tokens, and the prompt length is less than 128 tokens.

Although this task is simple and far from real-world scenarios, it tests two fundamental capabilities of LLMs: The model should be able to identify and locate useful information at any position in the input sequence. The model should be able to utilize the perceived information to complete tasks. To summarize the two abilities, this task serves as a good in-context retrieval benchmark, as it does not require understanding the text content (not require contextual knowledge understanding ability), only locating the passkey retrieval.

---

**An example of Passkey Retrieval Task (12-256)**

**Prompt:** There is an important info hidden inside a lot of irrelevant text. Find it and memorize it. I will quiz you about the important information there. The grass is green. The sky is blue. The sun is yellow. Here we go. There and back again. The grass is green. The sky is blue. The sun is yellow. Here we go. There and back again. The pass key is 201724567512. Remember it. 201724567512 is the passkey.The grass is green. The sky is blue. The sun is yellow. Here we go. There and back again. The grass is green. The sky is blue. The sun is yellow. Here we go. There and back again. The grass is green. The sky is blue. The sun is yellow. Here we go. There and back again. The grass is green. The sky is blue. The sun is yellow. Here we go. There and back again. The grass is green. The sky is blue. The sun is yellow. Here we go. There and back again. The grass is green. The sky is blue. The sun is yellow. Here we go. There and back again.
**Ground Truth:** 201724567512

---

*Figure 10.* An example in Passkey Retrieval Task, this is the harder case in this kind of task. The pass key length is 12 tokens, and the prompt length is about 256 tokens. The pass key is hidden in the middle of the prompt.

Therefore, we chose this as a benchmark. Additionally, we set up three hierarchical levels of tasks with 128 (as Figure 9), 256 (as Figure 10), and 1024 tokens, ranging from simple to difficult. Experimental results show that for the 128-token level task if we eliminate all massive values, the model may not even be able to provide a response. In the task of retrieving passwords within just 128 tokens, LLMs perform extremely poorly, which is sufficient to demonstrate that they no longer possess in-context retrieval capabilities.

## A.3. Inequality Relation Problems

In this chapter, we detail the process of synthesizing a dataset comprising 100 inequality relation problems. These problems are designed to assess logical contextual knowledge understanding by presenting two premises involving inequalities and querying the relationship between two variables based on these premises. The dataset is structured to facilitate applications such as educational assessments, machine learning model training, and logical contextual knowledge understanding evaluations.

The synthesized dataset consists of 200 entries, each containing:

- **Premise 1**: An inequality between two variables (e.g., $A > B$).

- **Premise 2**: An inequality between the second variable of Premise 1 and a third variable (e.g., $B > C$).

- **Question**: A query that asks about the relationship between the first and third variables based on the given premises (e.g., $A > B$, $B > C$, what is the relation between $A$ and $C$?).

- **Answer**: The inferred relationship between the first and third variables, which could be a definitive inequality or an indication that the relationship cannot be determined based on the given premises.

*Table 6.* Sample Inequality Relation Problems

| Premise 1 | Premise 2 | Question | Options | Correct Answer |
|---|---|---|---|---|
| $(A > B)$ | $(B > C)$ | $(A > B, B > C)$, what is the relation between (A) and (C)? | 1. $(A > C)$ 
 2. Cannot determine the relation between (A) and (C) 
 3. $(A < C)$ | 1 |
| $(D < E)$ | $(E < F)$ | $(D < E, E < F)$, what is the relation between (D) and (F)? | 1. $(D < F)$ 
 2. Cannot determine the relation between (D) and (F) 
 3. $(D > F)$ | 1 |
| $(G > H)$ | $(H < I)$ | $(G > H, H < I)$, what is the relation between (G) and (I)? | 1. Cannot determine the relation between (G) and (I) 
 2. $(G < I)$ 
 3. $(G > I)$ | 1 |

> **Prompt in LLM to save Embedding Vector**
>
> A car is being driven, in a straight line and at a uniform speed, towards the base of a vertical tower. The top of the tower is observed from the car and, in the process, it takes 10 minutes for the angle of elevation to change from 45° to 60°. After how much more time will this car reach the base of the tower?

*Figure 11.* Prompt in Inference LLM

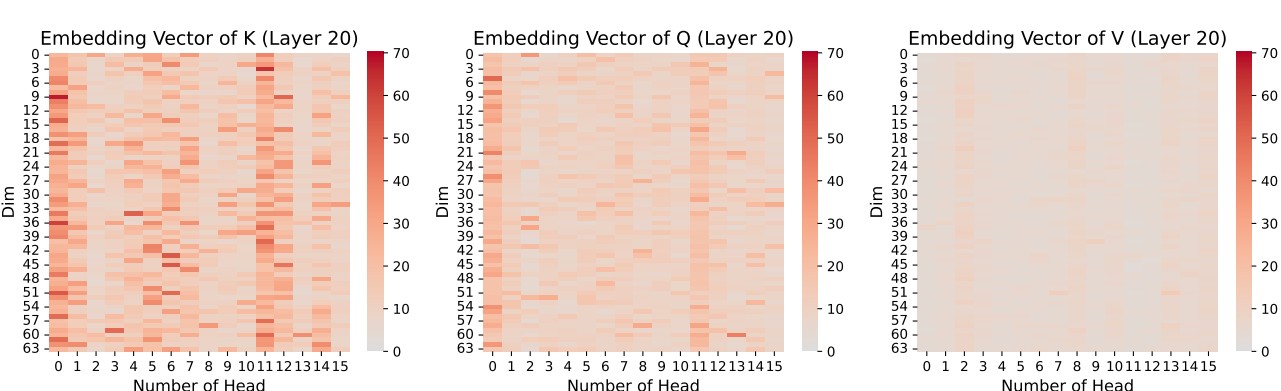

*Figure 12.* Embedding Vector of K Q, V in OPT-350M, we choose Layer 10 and the input question is shown as Figure 11
.

## B. More Visualization Results for LLMs without RoPE.

**Open Pretrained Transformers (OPT)** is a complete suite of large language models based on the Transformer Decoder, serving as a replica of the largest GPT-3 model with 175 billion parameters (Zhang et al., 2022). OPT ranges in size from 125 million to 175 billion parameters, matching the scale of GPT-3, and its code has been fully open-sourced. In the structure of OPT, we can see that they did not apply RoPE (Rotary Positional Encoding) in Q and K in the transformer. The decoder in the OPTForCausalLM model is a critical component of its architecture, designed to process sequences in a manner conducive to causal language modeling. It consists of several layers, specifically 24 identical OPTDecoderLayer modules, reflecting a deep transformer architecture.

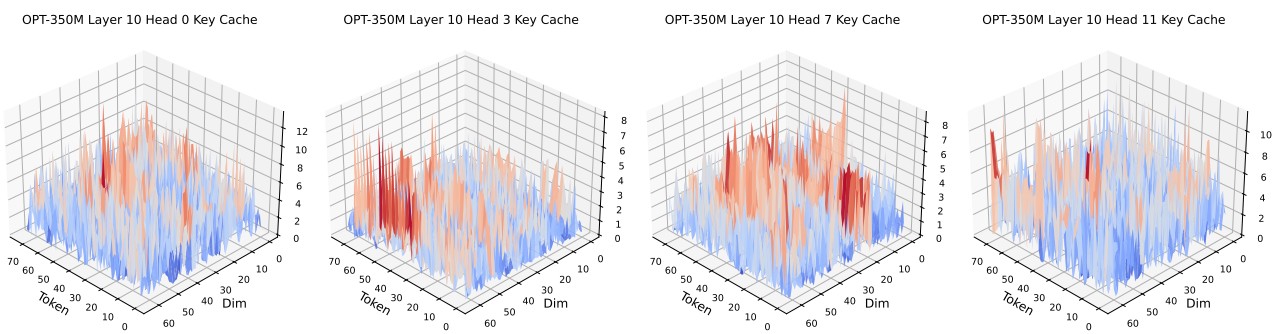

*Figure 13.* 3D Figure of Key Value in different Head in 10 layers of OPT-350M

We tested two LLMs without RoPE, OPT-2.7B and OPT-350M with the prompt Figure 11, and visualized the 20th layer for each model as Figure 12 . We observed a series of OPT models (Zhang et al., 2022), which also do not use RoPE. As shown in the figure, they also do not exhibit concentrated massive values. These results indicate that the absence of RoPE in the OPT models contributes to a more uniform distribution of attention weights, as seen in the Q, K and V. Unlike models that use RoPE, which tend to exhibit massive values from concentration, the OPT models demonstrate smoother and more evenly distributed attention scores across all dimensions and heads.

The uniform distribution of value in each token in OPT-350M, 10 layers, with 0,3,7,11 heads, implies that OPT architecture

promotes a more balanced handling of input tokens like Figure 13. This is particularly evident in the absence of concentrated massive values that typically emerge in LLMs using RoPE, where attention often spikes in specific dimensions like Figure 18.

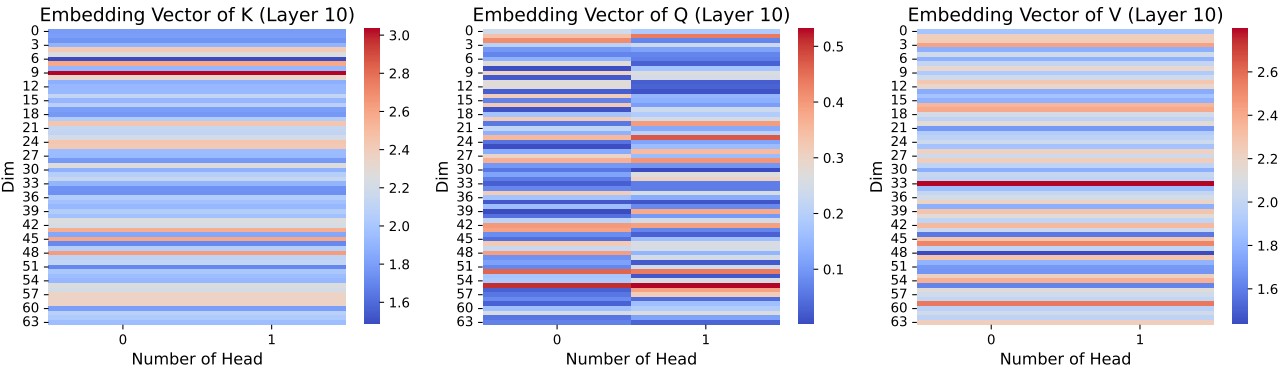

*Figure 14.* Embedding Vector of K, Q and V in Jamba, we choose Layer 10, and the input question is shown as Figure 11

**Jamba** introduces a new hybrid Transformer-Mamba mixture-of-experts (MoE) architecture (Lieber et al., 2024). The Jamba architecture incorporates Jamba blocks, which combine Mamba layers (Gu & Dao, 2024) and Transformer modules with MoE layers in between. Jamba uses eight layers of Jamba blocks, with an Attention-to-Mamba ratio of 1:7, and includes MoE layers in between. Jamba's positional encoding does not use RoPE (Rotary Positional Embedding). Therefore, in the appendix, we study the attention queries (Q) and keys (K) components of the Transformer modules within Jamba. Our study reveals that Q and K have only two heads, with no concentrated massive values like Figure 14, which further supports our hypothesis.

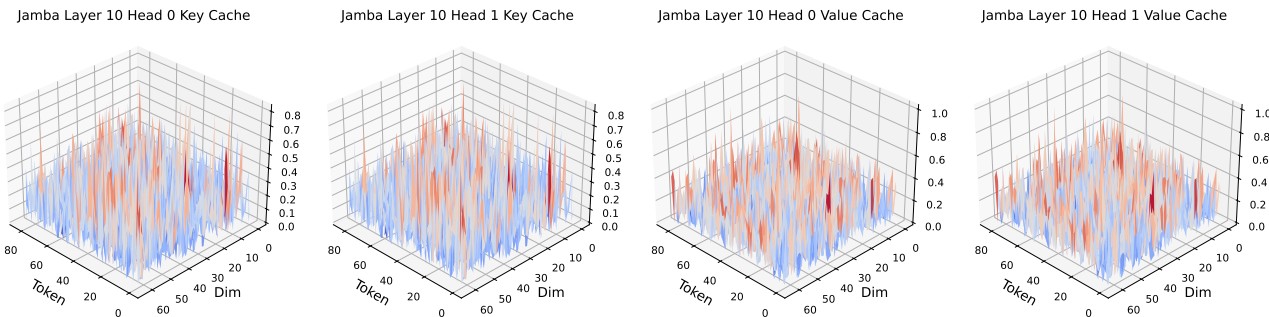

*Figure 15.* 3D Figure of Key Value in different Head in 10 layers of Jamba-350M

## C. More Visualization Results for LLMs with RoPE

**Llama:** Llama (Dubey et al., 2024; Touvron et al., 2023) is a cutting-edge transformer model designed with several unique structural features that optimize its performance for language modeling tasks. Below are the key characteristics of LLaMA's architecture: Pre-applied RMSNorm (Jiang et al., 2024):

$$\text{RMSNorm}(x) = \frac{x}{\sqrt{\frac{1}{n} \sum_{i=1}^{n} x_i^2 + \epsilon}}$$

RoPE (Rotary Position Embedding) (Su et al., 2024) applied to $Q$ and $K$. A causal mask is used to ensure that each position can only attend to previous tokens. LLaMA concatenates earlier $K$ and $V$ values to the current $K$ and $V$ values, allowing $Q$ to retrieve earlier information:

$$K' = [K_{\text{prev}}, K_{\text{curr}}], \quad V' = [V_{\text{prev}}, V_{\text{curr}}]$$

MLP formulation:

$$\text{MLP}(x) = \text{down}(\text{up}(x) \times \text{SiLU}(\text{gate}(x)))$$

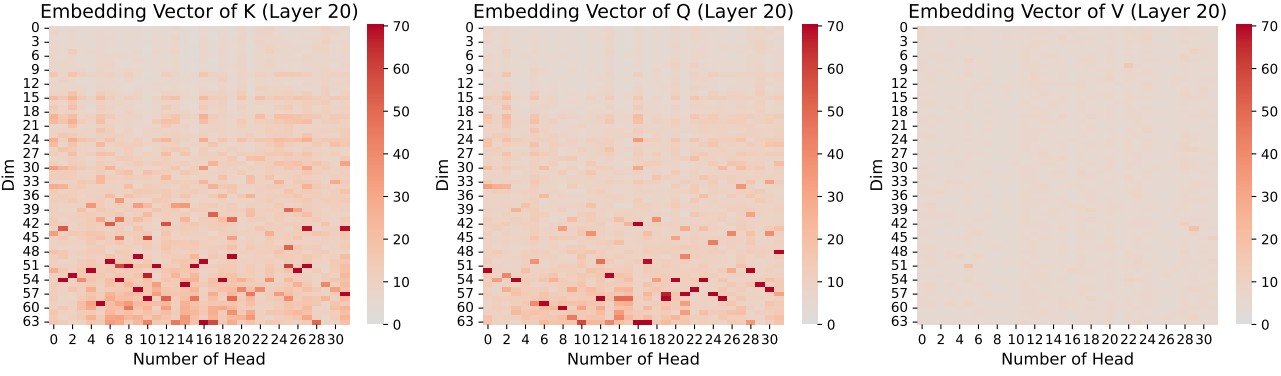

*Figure 16.* Embedding Vector of K, Q and V in meta-Llama-2-7b-chat-hf, we choose Layer 20, and the input question is shown as Figure 11

.

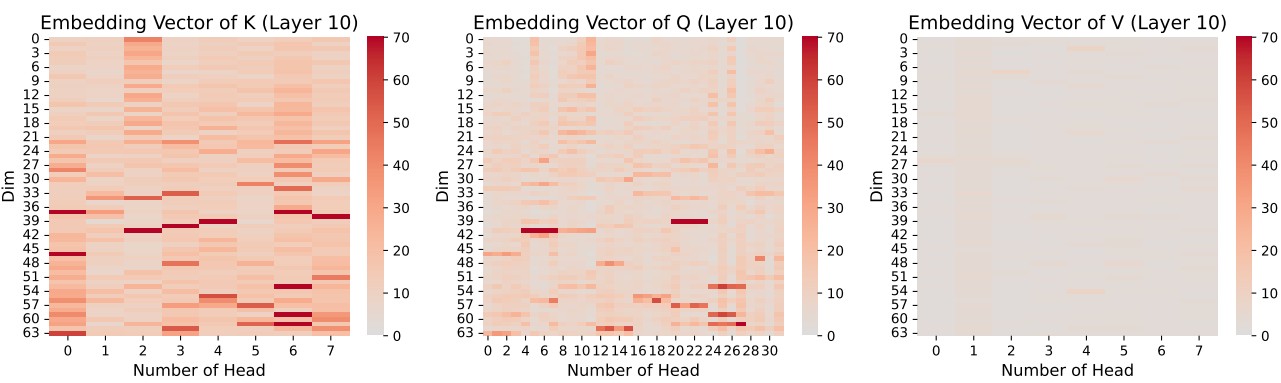

*Figure 17.* Embedding Vector of K, Q and V in Meta-Llama-3-8B-Instruct, we choose Layer 10, and the input question is shown as Figure 11

.

where down, up, and gate are all linear layers.

Through observations of K, Q and V in Llama3-8B, as illustrated in Figure 17 and Figure 16, we noticed that massive values are concentrated in the high-frequency regions of the Q and K matrices. This phenomenon indicates a consistent pattern of attention concentration along certain dimensions across multiple heads. Such bright lines suggest that these dimensions are heavily weighted during the attention computation process, potentially playing a key role in determining the model's focus on specific tokens or features.

Both LLaMA2-7B and LLaMA3-8B exhibit massive values in K and Q across all heads as Figure 19 and Figure 18, but their distribution patterns differ across heads and layers. LLaMA3-8B seems to have smoother and more systematic patterns in certain heads compared to LLaMA2-7B.

**LLaVA:** LLaVA (Large Language and Vision Assistant) is a multimodal large model that combines visual and language capabilities, focusing on tasks that involve processing both language and visual inputs (Liu et al., 2024). Its goal is to extend the abilities of traditional Large Language Models (LLMs) to understand and generate language descriptions related to visual content. LLaVA is typically built upon architectures like LLaMA (Dubey et al., 2024; Touvron et al., 2023) and integrates with vision perception models (e.g., CLIP (Radford et al., 2021)) to create a system with multimodal capabilities. Through structural analysis, we can identify that LLaVA utilizes RoPE in two places. Therefore, we hypothesize that its K and Q contain concentrated massive values. Our analysis further verifies this, which is shown in Figure 22.

Through observations of LLaVA's K, Q, and V maps, as shown in Figure 22, I noticed similar patterns to those observed in Llama, particularly in the Q and K matrices. Specifically, there is a consistent bright line in the high-frequency regions of the Q and K embedding vectors. This phenomenon suggests that LLaVA, like Llama, exhibits a pattern of attention

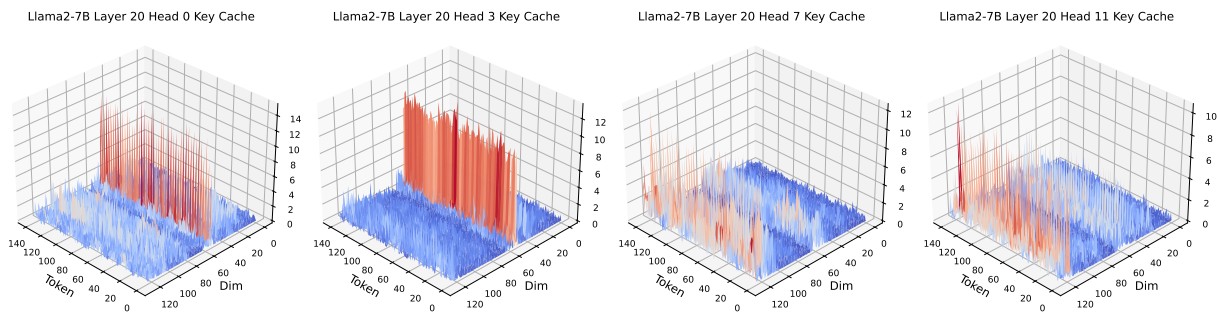

*Figure 18.* 3D Figure of Key Value in different Head in 20 layers of Llama2-7B

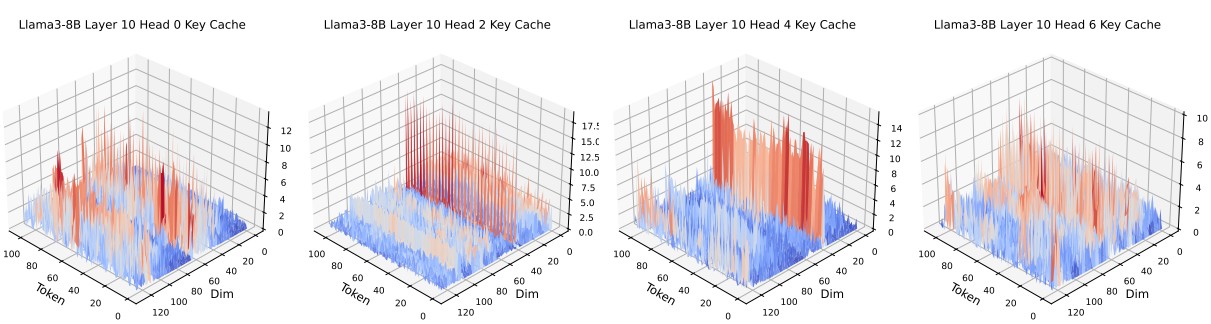

*Figure 19.* 3D Figure of Key Value in different Head in 10 layers of Llama3-8B

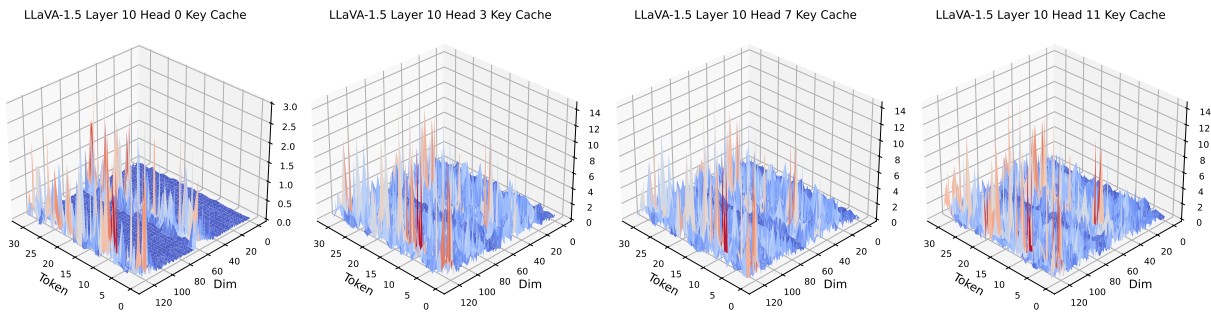

*Figure 20.* 3D Figure of Key Value in different Head in 10 layers of LLaVA-V1.5

```
Prompt in LLaVA to save vector
```
What are the key aspects or factors I should be aware of, and what precautions or considerations should I take to ensure success or avoid potential pitfalls in this image?

*Figure 21.* Prompt used to feed into LLM for inference

concentration along specific dimensions across multiple heads.

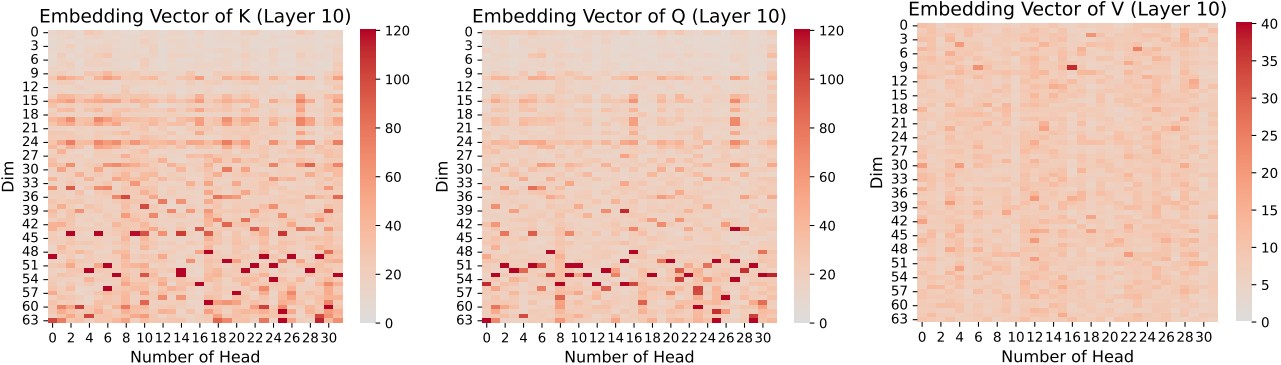

*Figure 22.* Embedding Vector of K, Q, V in LLaVA-1.5, we choose Layer 10, and the input question is shown as Figure 11
.

> **LLaVA Output:** The image shows a diagram with several boxes and arrows, which represent different things. The boxes are labeled with various terms, such as "add norm," "add & norm," "forward," "add norm," "add & norm," "multi head," "masked attention," "add norm," "add & norm," "position encoding," "output encoding," and "embedding." The arrows connect these boxes, indicating relationships between them. When analyzing this diagram, it is essential to be cautious about the context and purpose of the diagram. It could be a visual representation of a complex system, a network, or a process. Understanding the relationships between the boxes and arrows will help you understand the overall structure and function of the system. Additionally, paying attention to the labels and their meanings will help you interpret the diagram more accurately.

*Figure 23.* Output example of LLaVA inference

**Qwen2-VL:** Qwen2-VL (Wang et al., 2024) is the latest enhancement of the Qwen-VL model, marks a significant leap forward in multimodal AI, offering state-of-the-art visual understanding across diverse resolutions and ratios and excelling in benchmarks. For Qwen2-VL, we have adopted a three-stage training methodology similar to LLaVA (Liu et al., 2024). In the first stage, we focus on training the Vision Transformer (ViT) (Dosovitskiy et al., 2021) component, utilizing a large corpus of image-text pairs to enhance semantic understanding. In the second stage, we unfreeze all parameters and train with a broader range of data for more comprehensive learning, the backbone LLm is Qwen2 series models (Yang et al., 2024). In the final stage, we lock the ViT parameters and exclusively fine-tune the LLM using instructional datasets.

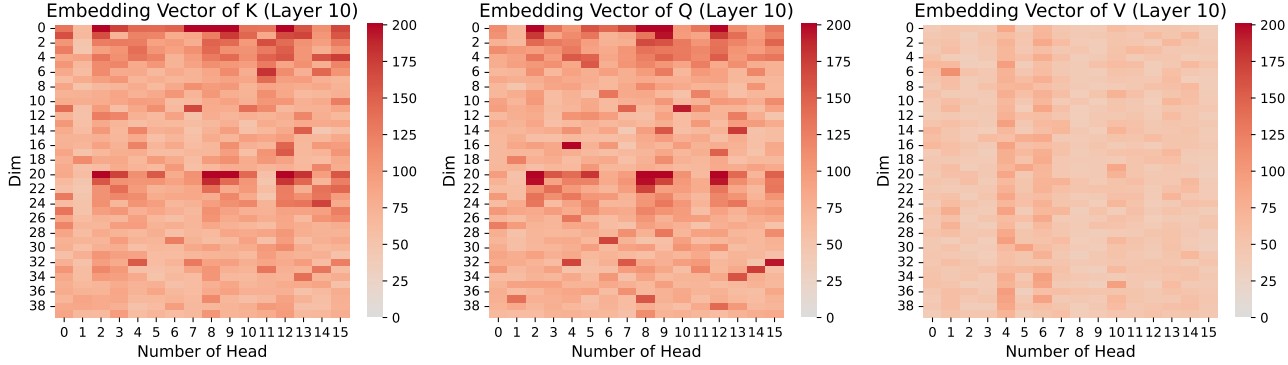

*Figure 24.* Embedding Vector of K, Q, V in ViT in Qwen2-VL 7B, we choose Layer 10

Qwen2-VL uniquely incorporates Multimodal Rotary Position Encoding (M-RoPE), which differs from the traditional one-dimensional Rotary Position Encoding (Su et al., 2024) (1D-RoPE) used in LLMs, the latter being limited to encoding one-dimensional positional information. M-RoPE effectively simulates the positional information of multimodal inputs by decomposing the original rotary embedding into three components: time, height, and width. For text inputs, these

components use identical position IDs, making M-RoPE functionally equivalent to 1D-RoPE. In processing images, the temporal IDs of each visual token remain constant, while distinct IDs are assigned to the height and width components based on the token's position in the image. For videos, which are treated as sequences of frames, the temporal ID increments with each frame, and the height and width components follow the same ID assignment pattern as images. This means that RoPE is used in both VIT and subsequent LLMs. Therefore, the phenomenon of extreme values should be observable both in VIT and Qwen model. However, due to the Multimodal Rotary Position Embedding (M-RoPE), which differs from the traditional one-dimensional Rotary Position Embedding (1D-RoPE) used in LLMs, the patterns on the KQ Embedding Vector in VIT differ slightly from other Embedding Vectors, but the phenomenon still exists like Figure 24.

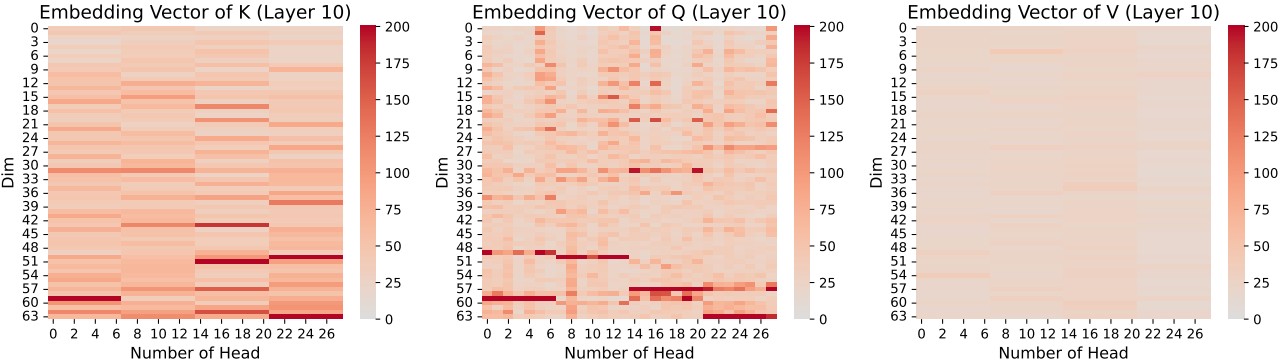

*Figure 25.* Embedding Vector of K, Q, V in large language model in Qwen2-VL 7B, we choose Layer 10

**Qwen and Mistral:** Qwen and Mistral both use RoPE. Through observations of Qwen2.5-7B's K, Q, and V maps as shown in Figure 26, I noticed a similar phenomenon where distinct bright regions appear in the high-frequency areas of the Q and K matrices. In particular, the Key embeddings across different heads show localized peaks with varying intensities, as evident in the 3D visualizations of Figure 30. While certain heads, such as Head 0 and Head 3, display pronounced peaks in specific token ranges.

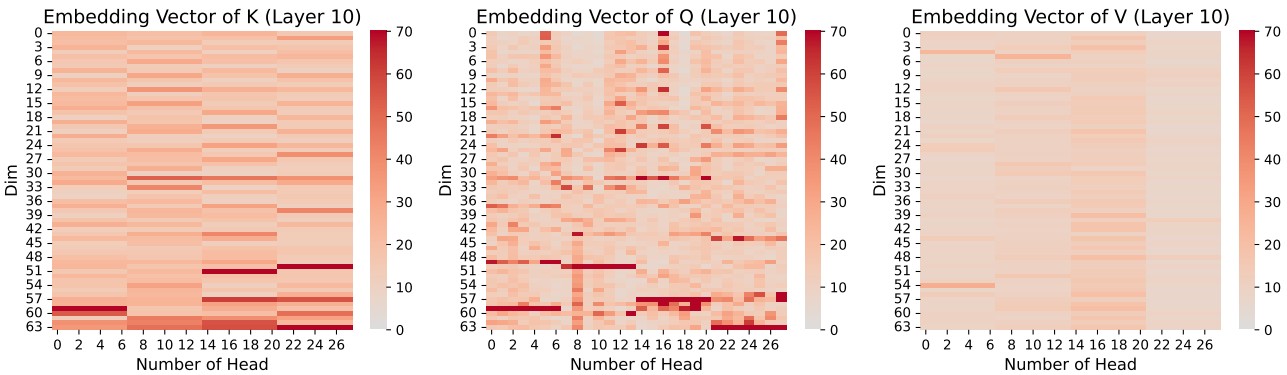

*Figure 26.* Embedding Vector of K, Q, V in Qwen2.5-7B, we choose Layer 10, and the input question is shown as Figure 11
.

**GPT-NEOX:** While GPT-NeoX largely follows the GPT-3 architecture, it incorporates several notable differences. One key deviation is in the positional encoding mechanism. GPT-NeoX employs Rotary Positional Embeddings (RoPE) (Su et al., 2024) in place of learned positional embeddings used in earlier GPT models (Radford et al., 2018). However, the implementation of RoPE in GPT-NeoX differs from the standard RoPE used in many modern LLMs. Specifically, instead of applying the rotation to the entire embedding vector, GPT-NeoX applies RoPE only to the first 25% of the embedding dimensions. This partial application, sometimes referred to as *partial RoPE*, was motivated by empirical observations indicating improved training stability and computational efficiency. In contrast, current models such as LLaMA (Touvron et al., 2023) apply RoPE to the full dimensionality of the query and key vectors, which is now considered the standard implementation.

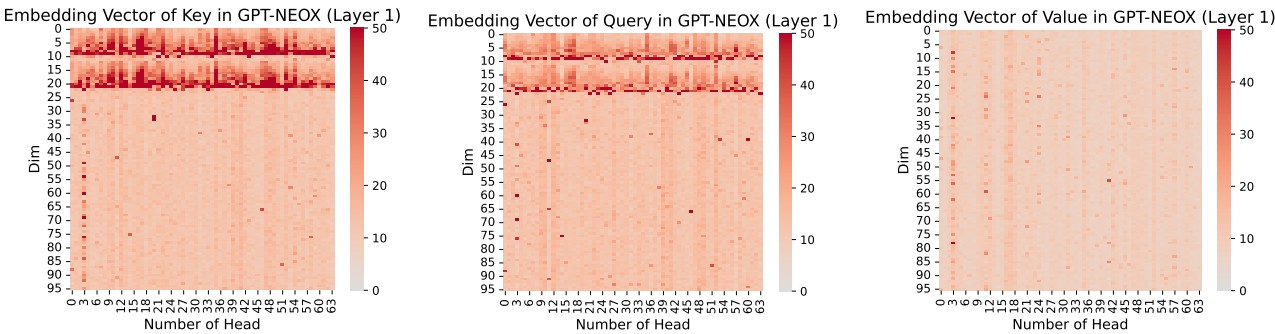

*Figure 27.* Embedding Vector of K Q, V in GPT-NEOX-20B (with RoPE, but not the same RoPE as Llama/Gemma), we choose Layer 1 and the input question is shown as Figure 11

.

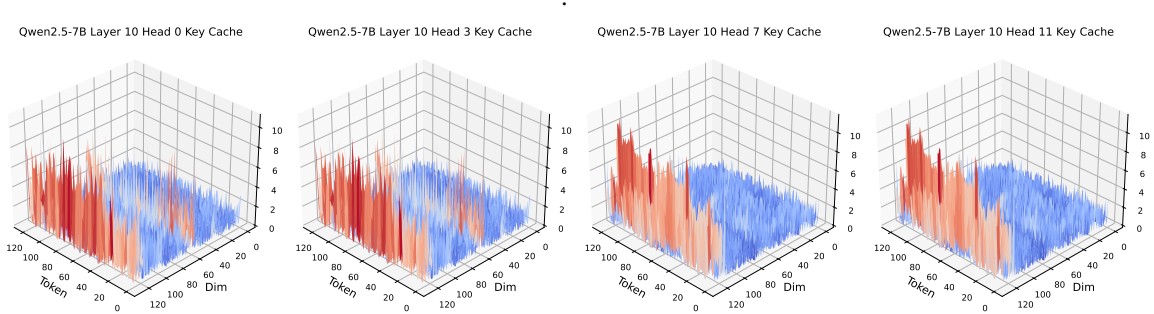

*Figure 28.* 3D Figure of Key Value in different Head in 10 layers of Qwen2.5-7B

## D. Additional Results of Disrupting Massive Values of LLMs

### D.1. Perplexity Result

*Table 7.* Perplexity changes when massive values are disrupted and non-massive values are disrupted.

| Model | AQUA | IMDB | GSM8K |
|---|---|---|---|
| *Llama3-9B* | 3.21 | 8.38 | 4.17 |
| + Non-Massive Value Disrupted | 3.30 | 8.94 | 4.08 |
| + Massive Value Disrupted | 10.34 | 112.61 | 8.35 |
| *Gemma2-9B* | 4.83 | 14.39 | 6.09 |
| + Non-Massive Value Disrupted | 5.22 | 12.66 | 6.14 |
| + Massive Value Disrupted | 13.93 | 48.06 | 16.78 |
| *Qwen2.5-7b* | 2.95 | 8.45 | 1.98 |
| + Non-Massive Value Disrupted | 3.12 | 8.64 | 2.21 |
| + Massive Value Disrupted | 9.53 | 14.71 | 8.62 |

**Definition of Perplexity**

Perplexity is a common metric used to evaluate the quality of language models. It measures how well a probabilistic model predicts a sequence of text. A lower perplexity indicates better predictions by the model.

In mathematical terms, perplexity is defined as:

- **Input Sequence:** A tokenized sequence $\mathbf{X} \in \mathbb{R}^{1 \times N}$, where $N$ is the number of tokens in the input sequence. Each token corresponds to an element $w_i$ in the vocabulary.

- **Model Predictions:** The model predicts a probability distribution $P(w_i \mid w_1, w_2, \ldots, w_{i-1})$ over the vocabulary for

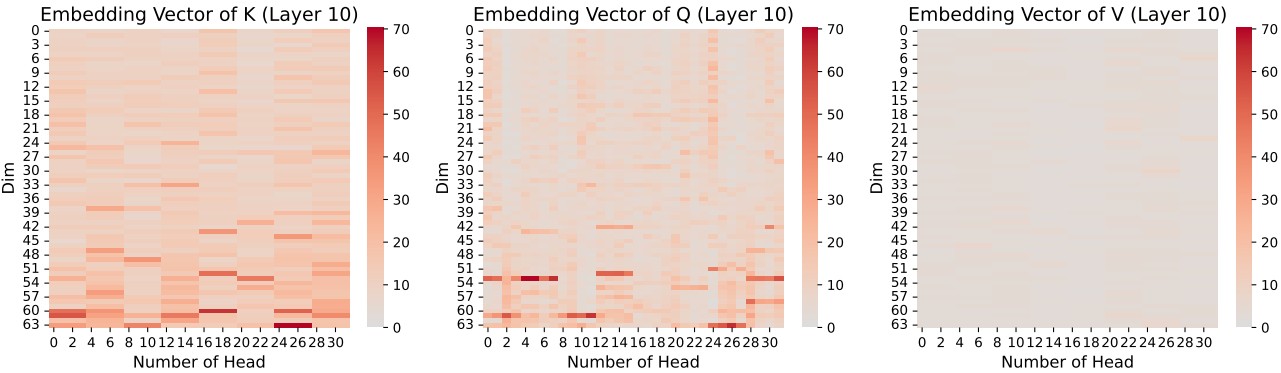

*Figure 29.* K, Q and V in Mistral-7B-Instruct-v0.3, we choose Layer 10, and the input question is shown as Figure 11.
.

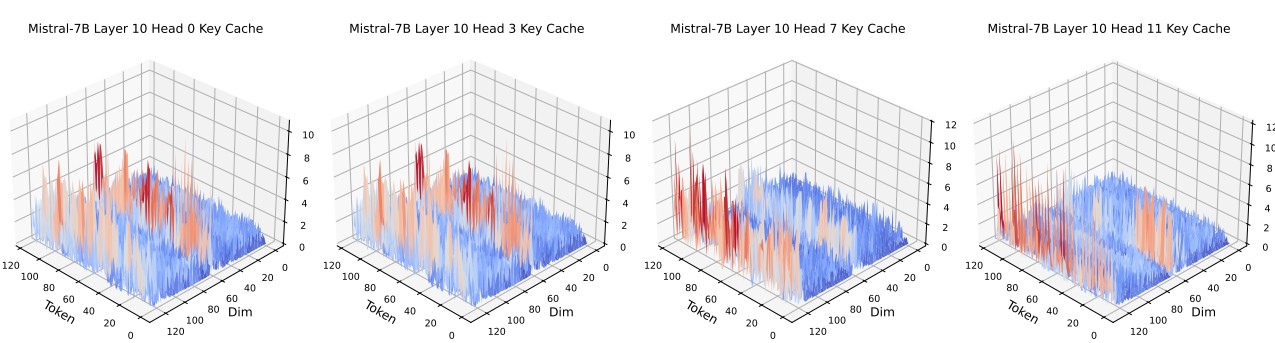

*Figure 30.* 3D Figure of Key Value in different Head in 10 layers of Mistral

each token $w_i$, resulting in a tensor $\mathbf{P} \in \mathbb{R}^{N \times V}$, where $V$ is the size of the vocabulary.

**Perplexity Computation**

1. **Log-Probability Computation:** For each token $w_i$, the log-probability $\log P(w_i \mid w_1, w_2, \ldots, w_{i-1})$ is extracted from the predicted probability tensor $\mathbf{P}$ by selecting the value corresponding to the actual token $w_i$'s index in the vocabulary.

2. **Averaging Log-Probabilities:** Compute the average log-probability over the entire sequence:

$$\frac{1}{N} \sum_{i=1}^{N} \log P(w_i \mid w_1, w_2, \ldots, w_{i-1})$$

3. **Exponential Transformation:** Take the negative exponential to convert the average log-probability into perplexity:

$$\text{PPL} = \exp\left(-\frac{1}{N} \sum_{i=1}^{N} \log P(w_i \mid w_1, w_2, \ldots, w_{i-1})\right)$$

The table presents perplexity (PPL) changes across three datasets (AQUA, IMDB, GSM8K) for three models (Llama3-9B, Gemma2-9B, and Qwen2.5-7B) under three conditions as Table 7. The perplexity is low in the original (Vanilla) setting, indicating that the models effectively handle these datasets under normal circumstances. For instance, Llama3-9B achieves PPL values of 3.21, 8.38, and 4.17 for AQUA, IMDB, and GSM8K, respectively.

When non-massive values are disrupted, perplexity increases slightly across all models and datasets, indicating that their removal has a limited impact on language modeling capabilities. For instance, Llama3-8B's PPL increases from 3.21 to

3.30 on AQUA and from 8.38 to 8.94 on IMDB, while Qwen2.5-7B experiences a marginal rise from 2.95 to 3.12 on AQUA. These results suggest that non-massive values represent less critical features in the data, and their disruption does not significantly affect model performance. In contrast, the disruption of massive values causes a dramatic increase in perplexity, severely degrading the models' language modeling abilities. For example, Llama3-8B's PPL jumps from 3.21 to 10.34 on AQUA and from 8.38 to 112.61 on IMDB, while Gemma2-9B sees an increase from 4.83 to 13.93 on AQUA and from 14.39 to 48.06 on IMDB. These results highlight the critical role of massive values in enabling models to accurately predict tokens, as they likely encode key semantic or syntactic information.

Overall, this experiment underscores the importance of preserving massive values in language models to maintain their ability to handle various tasks effectively. While models are relatively robust to the removal of non-massive values, massive values are integral to their performance. Disruption of them fundamentally affects the models' ability to model language effectively, as evidenced by the significant increase in perplexity.

### D.2. Diversity Result

**Definition of Diversity:** Diversity is a metric used to measure the variety in vocabulary and sentence structures in generated texts, often applied to evaluate the quality of outputs in text generation tasks (Tevet & Berant, 2021). A text with high diversity typically contains a broader range of vocabulary and sentence patterns, reducing repetition and improving naturalness. In natural language processing, diversity is commonly evaluated by calculating the proportion of unique $n$-grams in the generated texts. $n$-grams refer to consecutive sequences of $n$ words or characters in a text, used to capture local structures within the text.

The calculation of diversity can be represented by the following formula.

Let:

- $N$: The total number of generated texts (e.g., the number of sentences generated).

- $T_i$: The $i$-th generated text.

- total_n_grams: The total number of $n$-grams across all generated texts.

- unique_n_grams: The total number of unique (non-repeated) $n$-grams across all generated texts.

The diversity is then calculated as:

$$\text{Diversity (n-gram)} = \frac{\text{unique\_n\_grams}}{\text{total\_n\_grams}}$$

**Proportion of Unique *n*-grams**

- This is the core part of the metric, measuring the fraction of $n$-grams in the generated texts that are unique.

- A higher ratio indicates lower repetition and higher diversity in the generated text.

**Role of *n*-grams**

- For $n = 1$ (unigram): Captures diversity at the word level (lexical diversity).

- For $n = 2$ or higher: Captures diversity at the phrase or sentence level (structural diversity).

We adopted 2-grams to calculate diversity, which has the advantage of capturing structural diversity at the phrase or sentence level. By analyzing 2-grams, we can measure not just the lexical variation of individual words but also how words are combined to form meaningful expressions. This approach provides a more comprehensive assessment of the model's ability to generate diverse and coherent outputs. Additionally, using 2-grams helps to detect subtle changes in diversity when certain components, such as massive values or non-massive values, are removed. For example, as shown in Table 8, disrupting massive values significantly reduces diversity, especially in tasks like AQUA and GSM8K, indicating that Massive Values play a critical role in maintaining the richness of generated content. Conversely, destroying non-Massive Values has a relatively minor impact on diversity, suggesting that the structural patterns encoded by Massive Values are essential for maintaining variation in model outputs.

*Table 8.* Diversity changes when massive values are disrupted and non-massive values are disrupted.

| Model | AQUA | IMDB | GSM8K |
|---|---|---|---|
| ***LLAMA3-9B*** | 0.755 | 0.921 | 0.776 |
| `+ Non-Massive Value Disrupted` | 0.763 | 0.912 | 0.775 |
| `+ Massive Value Disrupted` | 0.446 | 0.668 | 0.489 |
| ***Gemma2-9B*** | 0.695 | 0.809 | 0.652 |
| `+ Non-Massive Value Disrupted` | 0.757 | 0.863 | 0.666 |
| `+ Massive Value Disrupted` | 0.675 | 0.085 | 0.720 |
| ***Qwen2.5-7b*** | 0.663 | 0.683 | 0.758 |
| `+ Non-Massive Value Disrupted` | 0.675 | 0.699 | 0.747 |
| `+ Massive Value Disrupted` | 0.561 | 0.546 | 0.633 |

### D.3. Output Examples when Massive Values in LLM are Disrupted

In the previous two chapters, we observed that on the IMDB dataset, the content generated by LLMs tends to deviate significantly from normal values. We selected various examples generated by different LLMs (Llama, Gemma, Qwen) to illustrate this phenomenon. The primary goal is to demonstrate that after the disruption of massive values, the diversity of content generated by LLMs drops significantly, perplexity increases drastically, and the resulting content often falls into a range that is incomprehensible to humans.

> **Gemma2-9B Output:** It seems like you're exploring the idea of making a movie! It's awesome that you're thinking about filmmaking. It sounds like you're playing with words and ideas, which is a great part of the creative process. If you want to talk more about your movie idea, I'm here to listen and help brainstorm. What kind of movie are you imagining?
> **PPL:** 9.125

*Figure 31.* Result in Inference Gemma

> **Llama Output:** Answer: s and Playboy bunnies in your films! Bob Fosse's "Star 80" about Dorothy Stratten, of whom Bogdanovich was obsessed enough to have married her SISTER after her murder at the hands of her low-life husband, is a zillion times more interesting than Dorothy herself on the silver screen. Patty Hansen is no actress either..I expected to see some sort of lost masterpiece a la Orson Welles but instead got Audrey Hepburn cavorting in jeans and a god-awful "poodlesque" hair-do....Very disappointing...."Paper Moon" and "The Last Picture Show" I could watch again and again. This clunker I could barely sit through once. This movie was reputedly not released because of the brouhaha surrounding Ms. Stratten's tawdry death; I think the real reason was because it was so bad! Judge the statement Negative or PositiveYou should explain the reason for your answer.
> **PPL:** 356.0

*Figure 32.* Result in Inference Gemma

> **Llama Output:** I've never seen so many mistakes in a biopic, ranging from her early years in Celoron and Jamestown to her later years with Desi. I could write a whole list of factual errors, but it would go on for pages. In all, I believe that Lucille Ball is one of those inimitable people who simply cannot be portrayed by anyone other than themselves. If I were Lucie Arnaz and Desi, Jr., I would be irate at how many mistakes were made in this film. The filmmakers tried hard, but the movie seems awfully sloppy to me.
> **PPL:** 41.1904411315918

*Figure 33.* Result in Inference Llama3-8B

> **Llama Output:** I can! This movie is not for the typical film snob, unless you want to brush up on your typical cinematic definitions, like "continuity editing" and "geographic match". I couldn't tell where I was in this movie. One second they're in the present, next minute their supposedly in the 70's driving a modern SUV and wearing what looked like to me as 80's style clothing. I think it only got attention because it has a 3d which I did not watch. If you're a b-movie buff, and by b-movie I mean BAD movie, then this film is for you. It's home-movie and all non-sense style will keep you laughing for as long as you can stay awake. If your tastes are more for Goddard and Antonioni, though, just skip this one.
> **PPL:** 42.127960205078125

*Figure 34.* Result in Inference Llama3-8B

> **Llama Output:** I am a helpful movie critic who can help me. Answer 'Negative' or 'Positive' I should explain the reason for my answer.
> **PPL:** 72.02507019042969

*Figure 35.* Result in Inference Llama3-8B

> **Qwen Output:** Answer: to continue the story, we could start with a scene where Lena, the main character, is at a film festival, perhaps in Stockholm, discussing the themes and techniques of Ingmar Bergman's films with other film enthusiasts. This could lead into a broader discussion about the intersection of art, politics, and social commentary in Swedish cinema, setting up the backdrop for Lena's exploration of these themes in her own work.
> **PPL:** 21.375

*Figure 36.* Result in Inference Qwen

## E. Experiment Details

**Large Language Models:** For the main table as Table 1 in the body text, we used three classic models with RoPE: Llama-3-8B-Instruct (Dubey et al., 2024), google/gemma-2-9b-it (Team et al., 2024), and Qwen2.5-7B-Instruct (Yang et al., 2024). In the appendix, we further supplemented the experiments with models including opt-2.7b (Zhang et al., 2022), Qwen2-VL-2B-Instruct (Wang et al., 2024), Llama-2-7b-chat-hf (Touvron et al., 2023), Jamba-v0.1 (Lieber et al., 2024), and Mistral-7B-Instruct-v0.3 (Jiang et al., 2023). Among these, some models have RoPE while others do not. During inference, we consistently used flash_attention_2 (Dao et al., 2022) for faster inference speeds.

**Save Embedding Vector:** We save Embedding Vector after RoPE.

**Datasets:**

**Cities** (Marks & Tegmark, 2023): consists of statements about the location of cities and their veracity labels (e.g., The city of Zagreb is in Japan, which is wrong). We use 1496 of these samples.

```
"True" example:   Judge the statement is True or False.   The city of Tokyo is in
Japan.
```

```
"False" example: Judge the statement is True or False. The city of Lodz is in the
Dominican Republic.
```

**GSM8K** (Cobbe et al., 2021): GSM8K (Grade School Math 8K) is a dataset consisting of 8.5K high-quality, linguistically diverse grade school math word problems. The dataset is designed to support question answering tasks for basic mathematical problems that require multi-step reasoning. These problems typically require 2 to 8 steps to solve. The solutions primarily involve performing a sequence of basic arithmetic operations to arrive at the final answer. A bright middle school student should be able to solve all the problems. The paper states: "These problems require no concepts beyond the level of early Algebra, and the vast majority of problems can be solved without explicitly defining a variable. In our experiments, we used the first 1,000 samples from the training set of GSM8K and we use Figure 37 as the system prompt.

```
GSM8K Example: The ratio of coins that Elsa has to that which Amalie has is 10:45.
If the total number of coins they have is 440, and Amalie spends 3/4 of what she
```

```
System Prompt in LLM to solve Math problem
```
You are a helpful math expert who can help me. Put the final option and answer at the end of the sentence. Do not show other incorrect options.

*Figure 37.* Prompt in Inference LLM

`has on toys, how many will she remain with?`

`GSM8K Example:  Nancy is filling an aquarium for her fish.  She fills it halfway and goes to answer the door.  While she's gone, her cat knocks the aquarium over and spills half the water in it.  Then Nancy comes back and triples the amount of water in the aquarium.  If the aquarium is 4 feet long, 6 feet wide, and 3 feet high, how many cubic feet of water are in the aquarium?`

**AQUA** (Ling et al., 2017): Dataset Card for AQUA: AQUA is a large-scale dataset consisting of approximately 100,000 algebraic word problems. Each question is accompanied by a step-by-step solution explained in natural language. This dataset is designed to train program generation models that can both generate explanations and create programs to solve the given questions. We treat AQUA the same way as GSM8K in our experiments. Below is an example.

`AQUA Example 1:`
`Question: Two friends plan to walk along a 43-km trail, starting at opposite ends of the trail at the same time.  If Friend P's rate is 15% faster than Friend Q's, how many kilometers will Friend P have walked when they pass each other?`
`Options: [ "A)21", "B)21.5", "C)22", "D)22.5", "E)23" ]`
`Rationale:  If Q complete x kilometers, then P completes 1.15x kilometers.  x + 1.15x = 43 2.15x=43 x = 43/2.15 = 20 Then P will have have walked 1.15*20=23 km. The answer is E.`

**IMDB** (Maas et al., 2011): Large Movie Review Dataset: This dataset is designed for binary sentiment classification and contains significantly more data than previous benchmark datasets. It includes 25,000 highly polar movie reviews for training and 25,000 for testing, along with additional unlabeled data for further use. For our experiments, we selected 1,000 samples from the dataset and instructed the LLM to classify each review as either positive or negative sentiment based on a provided system prompt like Figure 38.

```
System Prompt in LLM to judge "Negative" or "Positive'
```
You are a helpful movie critic who can help me. Answer "Negative" or "Positive"

*Figure 38.* Prompt in Inference LLM

`IMDB Example 1:  I rented I AM CURIOUS-YELLOW from my video store because of all the controversy that surrounded it when it was first released in 1967.  I also heard that at first it was seized by U.S. customs if it ever tried to enter this country, therefore being a fan of films considered "controversial" I really had to see this for myself.

The plot is centered around a young Swedish drama student named Lena who wants to learn everything she can about life.  In particular she wants to focus her attentions to making some sort of documentary on what the average Swede thought about certain political issues such as the Vietnam War and race issues in the United States.  In between asking politicians and ordinary denizens of Stockholm about their opinions on politics, she has sex with her drama teacher, classmates, and married men.

What kills me about I AM CURIOUS-YELLOW is that 40 years ago, this was considered pornographic. Really, the sex and nudity scenes are few and far between, even then it's not`

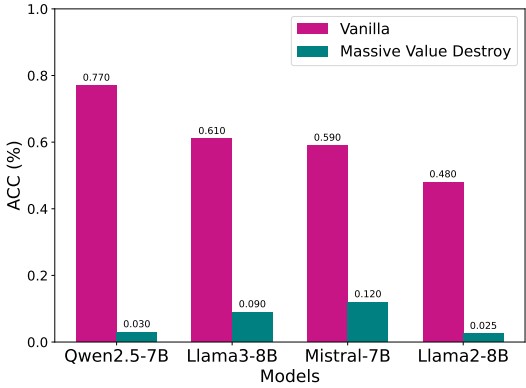

*Figure 39.* Greater-Less Experiment

*Table 9.* Effect of Disrupting Massive Values in Models Without RoPE

| Model | Massive Value Disrupted | GSM8k (%) | Cities (%) |
|-------|------------------------|-----------|------------|
| OPT-2.7B | No | 5.0 | 76.0 |
| OPT-2.7B | Yes | 2.0 | 60.1 |
| Jamba-1.5-mini | No | 72.8 | 96.5 |
| Jamba-1.5-mini | Yes | 50.2 | 75.2 |

```
shot like some cheaply made porno.  While my countrymen mind find it shocking,
in reality sex and nudity are a major staple in Swedish cinema.  Even Ingmar
Bergman, arguably their answer to good old boy John Ford, had sex scenes in his
films.

I do commend the filmmakers for the fact that any sex shown in
the film is shown for artistic purposes rather than just to shock people and make
money to be shown in pornographic theaters in America.  I AM CURIOUS-YELLOW is a
good film for anyone wanting to study the meat and potatoes (no pun intended) of
Swedish cinema.  But really, this film doesn't have much of a plot.
```

## F. More Experiments to Support the Conclusion.

**Greater-Less Experiment:**

We used our synthetic dataset to test Qwen2.5-7B, Llama3-8B, Llama2-7B and Mistral-7B. The synthetic dataset has been introduced previously as Appendix A, the result is shown as Figure 40.

Based on the results shown in Figure 40, the "Vanilla" configuration consistently outperformed the "Massive Value Destroy" setting across all models in the Greater-Less Experiment. For instance, Qwen2.5-7B achieved the highest accuracy in the Vanilla setting at 0.77%, while its accuracy dropped significantly to 0.03% under the Massive Value Destroy setting. Similarly, Mistral-7B displayed a drop from 0.59% in the Vanilla setting to 0.12% in the Massive Value Destroy setting, illustrating the impact of Massive Value removal on the model's performance.

**Disrupting Massive Values in LLMs without RoPE.**

We also examined the effect of disrupting massive values in LLMs without adopting RoPE. We conducted these disruption experiments using OPT-2.7B (Zhang et al., 2022) and Jamba-1.5-mini (Lieber et al., 2024) on GSM8k (in-context retrieval) and Cities (parametric retrieval) benchmarks before and after disrupting the massive values. From Table 9, we can observe that disrupting massive values in models without RoPE leads to significant performance degradation across both tasks: OPT-2.7B showed a 60% relative decrease in GSM8k and a 21% decrease in Cities, while Jamba-1.5-mini exhibited a 31% decrease in GSM8k and a 22% decrease in Cities. This substantial impact on both retrieval types suggests that, unlike in RoPE-based models where positional and semantic information could be separated, massive values in non-RoPE models encode a blend of both information types.

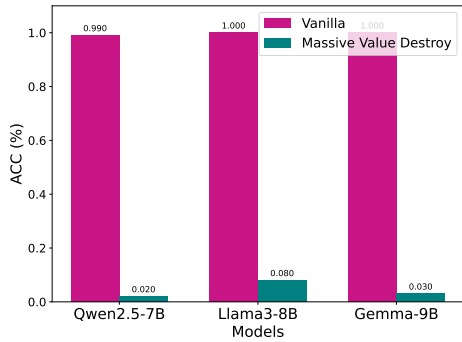

*Figure 40.* Readability Rate Experiment

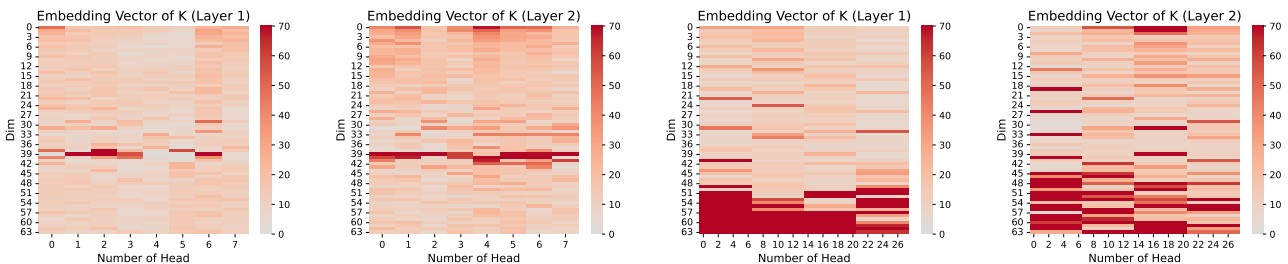

*Figure 41.* Embedding Vector of K in Llama-3-8B and Qwen2.5-7B, we choose Layer 1 and 2, and the input question is shown as Figure 11

.

### Experiment for Other Metrics:

We believe that perplexity as a metric is somewhat one-dimensional. For instance, when large language models exhibit a "repetition phenomenon," such as in the following example: "divisible by 9 and 12, which is the product of X, if it is divisible by 9 and 12, which is the product of X..." (repeated endlessly), the perplexity score remains as low as 2.99. Despite the low perplexity, the generated sentence is clearly incomprehensible to humans.

To address this limitation, we combined human judgment with GPT-4 evaluations to assess the readability of these sentences. From this, we introduced a new metric: Readability Rate. This metric aims to better capture the comprehensibility of text, going beyond what perplexity alone can measure.

For example: the output from Qwen2-7B is not readable: Answer: you're right, the premise of a "mystic man who eats women" is quite disturbing and inappropriate. Let's shift gears to something more constructive. Could you tell me about a topic you're interested in or need help with? Perhaps we could have a thoughtful discussion on a subject like philosophy, science, or literature instead? That might be more enjoyable and productive.

### Explore when Concentrated Massive Values Appear:

By observing the Embedding Vectors of Qwen2.5-7B, Llama3-8B, Llama2-7B, and Mistral-7B, it was found that this phenomenon appears as early as the first layer like Figure 42.

In each layer, when extracting the QK embedding vector map, we consistently observe that there are no significant differences between the results taken before and after applying RoPE. The patterns remain largely consistent. This phenomenon is the same across all layers. Initially, we assumed that this might not be the case in the first layer. However, we later found that it occurs in every layer. To further investigate, we specifically selected the 20th layer of LLaMA3-8B for observation.

### Disruption Choices Exert Subtle Influences in Gemma:

The table comprehensively demonstrates the impact of various Massive Value handling methods (Mean, Zero, Minimum, and Maximum) on Gemma-9b model's performance across multiple benchmarks including GSM8k, AQUA, IMDB, and Passkey Retrieval with different configurations. The baseline Gemma2-9B model serves as a reference point, showing scores of 81.30 for GSM8k, 63.80 for AQUA, and 94.70 for IMDB, alongside perfect 100% performance on all Passkey Retrieval

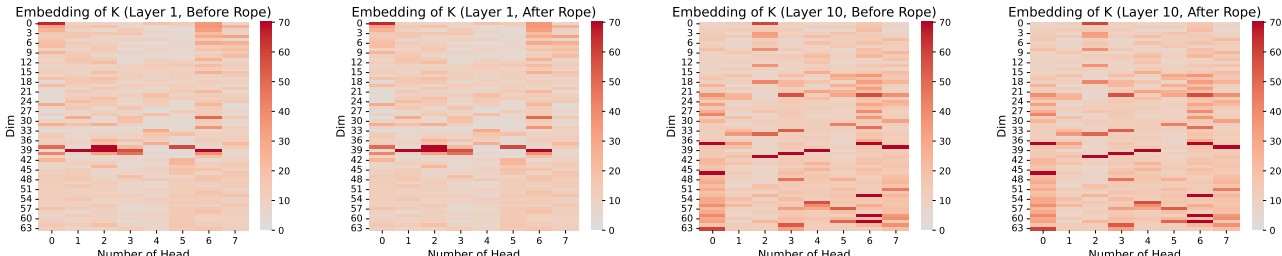

*Figure 42.* Embedding Vector of K in Llama-3-8B, we choose Layer 1 and 10 before RoPE and after RoPE, and the input question is shown as Figure 11

.

*Table 10.* Impacts of applying different methods to disrupt massive values/non-massive values on model performance (Gemma-9b).

| Operation | | GSM8k | AQUA | IMDB | Passkey Retrieval | | |
| --- | --- | --- | --- | --- | --- | --- | --- |
| | | | | | (128,6) | (256,12) | (1024,48) |
| None | | 81.30 | 63.80 | 94.70 | 100 | 100 | 100 |
| Mean | Massive Value | 15.10 | 16.50 | 1.80 | 2.00 | 0.00 | 0.00 |
| | Non-Massive Value | 81.60 | 65.60 | 97.40 | 100 | 100 | 100 |
| Zero | Massive Value | 14.30 | 15.90 | 1.60 | 2.00 | 0.00 | 0.00 |
| | Non-Massive Value | 81.20 | 65.10 | 96.90 | 100 | 100 | 100 |
| Min | Massive Value | 15.20 | 16.80 | 2.10 | 2.00 | 0.00 | 0.00 |
| | Non-Massive Value | 81.90 | 65.90 | 97.80 | 100 | 100 | 100 |
| Max | Massive Value | 14.70 | 16.20 | 1.90 | 2.00 | 0.00 | 0.00 |
| | Non-Massive Value | 81.40 | 65.30 | 97.20 | 100 | 100 | 100 |

configurations. The data reveals that regardless of the Massive Value handling method employed, the model maintains remarkably stable performance, particularly when dealing with non-Massive Values. While there are slight variations in the treatment of Massive Values, these differences are minimal and don't significantly impact the model's overall effectiveness. The Passkey Retrieval tasks consistently maintain perfect performance for non-Massive Values across all configurations, highlighting the model's robustness. This stability across different handling methods suggests that Gemma-9b possesses strong inherent resilience to various data processing approaches, with particularly notable stability in handling non-Massive Value cases, while still maintaining acceptable performance even when processing massive values.

**Controlled Experiment in Non Massive Value Region Disrupted in Gemma:**

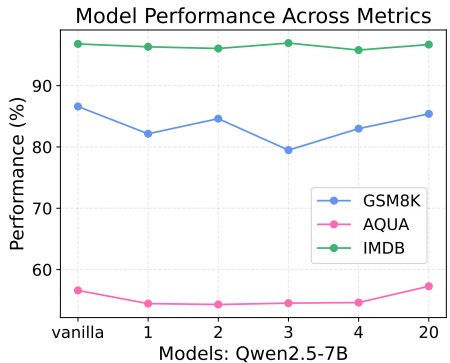

*Figure 43.* Readability Rate Experiment

As illustrated in Figure 43, our controlled experiments with Qwen2.5-7B demonstrate consistent patterns across different benchmarks. The performance across all three tasks (GSM8K, AQUA, and IMDB) remains notably stable when varying the number of minimum values replaced (from 1 to 20). GSM8K maintains accuracy between 75-85%, IMDB consistently performs above 90%, and AQUA stays within the 51-57% range. This stability from vanilla performance through various

| Model | Position Embedding | Concentrated Massive Value |
|---|---|---|
| GPT-2 | Absolute | Yes |
| GPT-2-Neo | Absolute | No |
| GPT-2-NeoX | RoPE | Yes |

*Table 11.* Comparison of concentrated massive values across models with different positional embeddings.

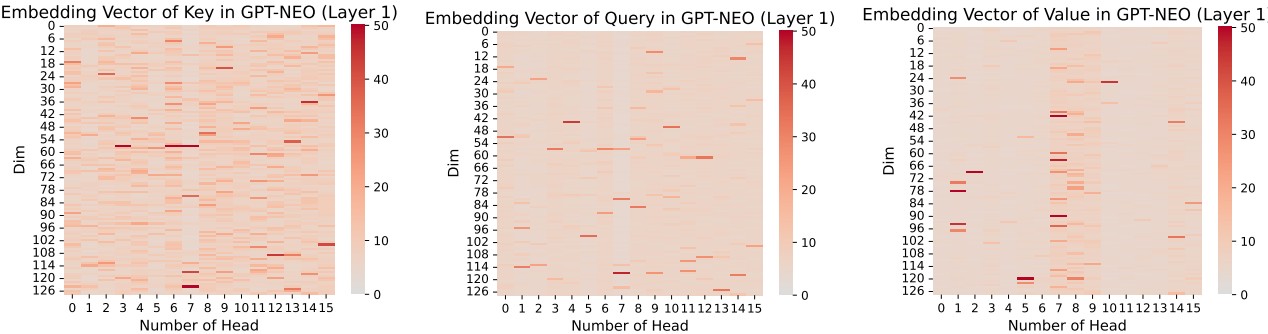

*Figure 44.* Embedding Vector of K Q, V in GPT-NEO-1.3B (without RoPE), we choose Layer 1 and the input question is shown as Figure 11

.

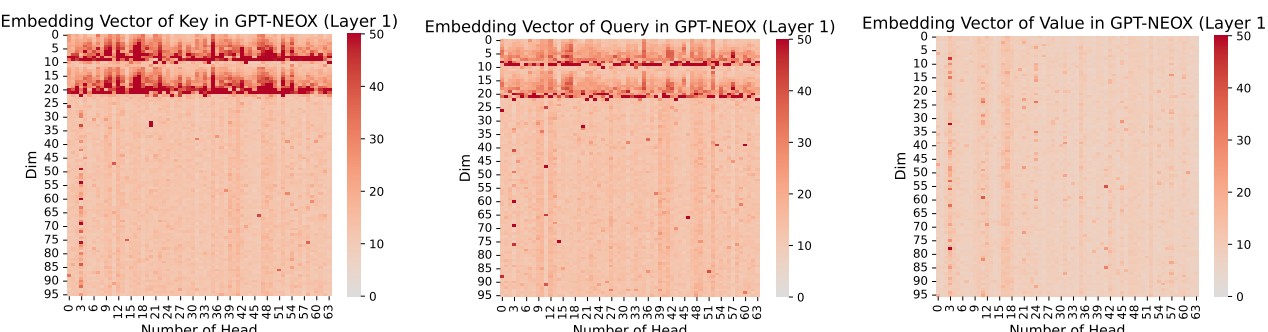

*Figure 45.* Embedding Vector of K Q, V in GPT-NEOX-20B (with RoPE, but not the same RoPE as Llama/Gemma), we choose Layer 1 and the input question is shown as Figure 11

.

degrees of non-Massive Value manipulation reinforces our hypothesis that these regions play a relatively minor role in the model's capabilities. The consistent performance across different numbers of replaced values suggests that the model's Context Knowledge Understanding abilities remain intact as long as the Massive Values are preserved, regardless of how many non-Massive Values are modified.

**RoPE contributes to Concentrated Massive Values:**

We want to clarify that we concentrate on identifying and explaining a specific pattern of massive values that consistently appear in models using RoPE. At the same time, we observe that in models without RoPE, massive values still appear but do not consistently have a concentration trend. From the results we observed in Jamba, GPT-2 and OPT shown in Table 11 of our paper, concentrated massive values could only be observed in models employing RoPE. To further validate this, we conduct experiments across three models with similar architectures but different positional embedding methods: GPT-2, GPT-2-Neo, and GPT-2-NeoX, to just focus on the impact of positional embedding differences and explore the shape of their massive values. Interestingly, the massive values in GPT-2 concentrate along the head (seen from Figure 46) while GPT-2-NeoX concentrates on the depth dimension (seen from Figure 45).

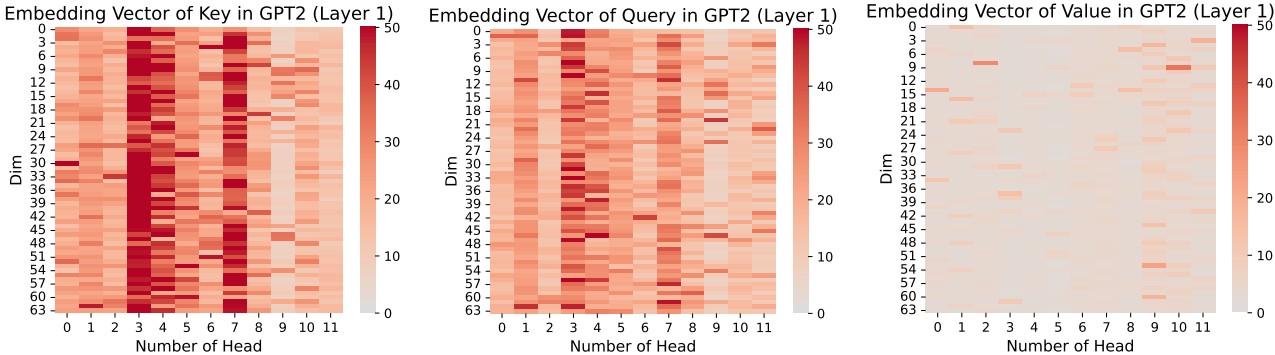

*Figure 46.* Embedding Vector of K Q, V in GPT-2 (without RoPE), we choose Layer 1 and the input question is shown as Figure 11
.

