# OpenReview forum: "Massive Values in Self-Attention Modules are the Key to Contextual Knowledge Understanding"
_ICML.cc/2025/Conference — ICML 2025 poster_

### Official Review · Reviewer_p2F4 · 2025-03-08

**Overall Recommendation:** 3

**Summary:**

The paper empirically observed that LLM with RoPE would have massive QK values, which has positive correlation to the knowledge understanding capability.
## updated after rebuttal

**Claims And Evidence:**

LLM with RoPE would have massive QA value pattern, which is responsible for knowledge understanding. The author conducts several experiments on several models to substantiate the claims.

**Essential References Not Discussed:**

Allen-Zhu and Li. (2024a) also highlighted the importance of RoPE, but the author didn’t discuss them.

**Experimental Designs Or Analyses:**

Allen-Zhu and Li. (2024a) also highlighted the importance of RoPE, but the author didn’t discuss them.

**Methods And Evaluation Criteria:**

The paper utilized different disruption strategies to LLM in different tasks, and validated through task-specific metric.

**Other Comments Or Suggestions:**

Allen-Zhu and Li. (2024a-b) also highlighted the importance of RoPE (see Figure 4 in Allen-Zhu and Li. (2024a)). Please provide a discussion.

**Other Strengths And Weaknesses:**

**Strengths**

1. The paper offers an interesting perspective on analyzing the RoPE’s role, and connects the LLM’s capability of knowledge understanding to massive values of QA.’
2. The logic is generally nice.

**Weakness**

1. The authors fail to employ the methods on knowledge editting / benign forgetting tasks based on the findings in Section 3.5, which is of the community’s great interest and might complement the contributions.
2. The paper only showed the relevance between massive QA value and the capability of knowledge understanding, without delving into why (e.g. how do massive QA values help LLM implement certain algorithms such as DP algorithms Allen-Zhu and Li. (2024a)).
3. The paper do not analyze the gradient-whether it is the bias of gradient descent that leads to massive QA values when adopting RoPE. Also, whether the massive QA value is necessary for LLM to obtain strong knowledge understanding capability, or it just would guide the LLM to reach some sub-optimal points in the highly-non convex optimization landscape by gradient methods (in terms of the considered objective goal)? That suggests, other PE mechanisms also have the potential to excel without the massive values patterns of QK, which might be more robust to certain poisoning attacks.
4. The best values or the most crucial ones across tasks should be highlighted in Table 1.

**Questions For Authors:**

1. Why there isn’t concat operation over Q in (5)?
2. I’m interested in two cases testing LLM + RoPE:  (i)“What is the capital of America: A; What is the capital of Japan:”. If the model can correctly answer “J”, denoting the model successfully processed the context to recognize that “capital” here means “capital letter”, would the model fail to do so if you adopt the disruption strategy of massive value? (ii) On the other hand, when the prompt is “What is the capital of America: Washinton DC; What is the capital of Japan:”, if you then adopt the disruption, what would the model response?
3. I don’t really understand your discussions under Hypothesis 1 paragraph. Is it the case that RoPE naturally creates some low-frequency regions around important positions, which causes the QA’s value concentrate on them and thus help improve contextual understanding?
4. In my intuition, semantic understanding and knowledge retrieval all depends on vector representation, could your results imply that knowledge retrieval depends more on the value matrices’ meaningful representation, while semantic understanding replies on both QK and V?

*Reference*

Allen-Zhu and Li. (2024a) Physics of Language Models: Part 1, Learning Hierarchical Language Structures.

Allen-Zhu and Li. (2024b) Physics of language models: Part 3.1, knowledge storage and extraction.

**Relation To Broader Scientific Literature:**

N/A

**Theoretical Claims:**

N/A

---

> ### Author Rebuttal · Authors · 2025-03-29
>
> > D1: Allen-Zhu and Li also highlighted the importance of RoPE. Please provide a discussion.
>
> Thank you for the excellent suggestion. We acknowledge that our current version does not yet include a discussion of Allen-Zhu and Li [1]. We appreciate the reviewer’s input and will include proper citations and debate in the next version.
>
> Their work provides valuable insights into how different position-encoding mechanisms affect LLM performance. While we share similar interests, our focus is distinct: they emphasize performance, whereas we analyze how RoPE leads to massive values concentrating on specific dimensions of the K/V embeddings and how these values are crucial for contextual knowledge understanding. We believe their findings complement ours and will make this connection more explicit in the next revision.
>
> > Q1: Why isn’t there a concat operation over Q in (5)?
>
> Thank you for the question. The absence of concatenation over the Query matrix Q in Equation (5) is intentional and follows the standard design in transformer-based LLM. During autoregressive generation, **Q is recomputed** at each decoding step for the current token; **K and V are cached** across time steps to avoid redundant computation [2]. Our formulation mirrors the **prefill-stage equations** used in prior work [2, 3], where concatenation is applied only to K and V (the cached tensors) but not to Q. Since the Query is computed afresh at each decoding step and is not cached, it does not require concatenation with past representations.
>
> > Q2: Can LLMs with RoPE still correctly interpret the meaning of "capital" in different contexts and respond accurately after applying disruptions?
>
> Thank you for the insightful question and illustrative examples. In case (i) What is the capital of America: A; What is the capital of Japan: without disruption, the model infers that “capital” refers to capital letters, responding with J. However, the model loses this contextual understanding under our massive value disruption and defaults to Factual knowledge recall, answering Tokyo.  In case (ii) “What is the capital of America: Washington DC; …” The context frames “capital” in a geopolitical sense. Thus, the model answers “Tokyo” in both settings, with or without disruption.
>
> We construct a 100-example test set in two formats to validate like the table below. Results consistently show that **disruption suppresses contextual understanding, forcing models to rely on memorized knowledge**. We will include the evaluation in the updated version.
>
>
> | Model  | Dataset Type                               | Undisrupted Acc | Disrupted Acc |
> |--------|--------------------------------------------|------------------|----------------|
> | Qwen   | Contextual Understanding (“capital of America: A,” expecting “J”) | 100              | 10             |
> | Qwen   | Factual Recall (“Washington DC”)     | 100              | 94             |
> | LLaMA  | Contextual Understanding                          | 100              | 10             |
> | LLaMA  | Factual Recall                              | 100              | 89             |
>
>
> >  Q3: Clarification on Hypothesis 1:
>
> Thank you for the thoughtful question. We clarify that RoPE naturally induces low-frequency components in the lower dimensions of Q and K, which tend to accumulate extreme values [4]. These values often correspond to semantically important tokens and enhance contextual understanding. In contrast, high-frequency components (in upper dimensions) encode fine-grained positional information, so the semantic information is weak in that area [4], so the massive value does not appear in such area. When we disrupt these massive value regions (e.g., through masking or perturbation), the model’s ability to capture semantic information degrades significantly, supporting our claim that RoPE critically shapes embedding for semantic reasoning.
>
>
> >  Q4: On the Role of QK vs. V in Semantic Understanding and Knowledge Retrieval
>
> We agree with the reviewer that semantic understanding and knowledge retrieval rely on vector representations. However, our work does not compare the relative roles of QK and V. Our focus is on massive value outliers in the QK embeddings, which lead to highly concentrated distributions due to RoPE position embedding. We show that removing these outliers impairs tasks requiring contextual understanding, such as math reasoning. Rather than making broad claims, we highlight a specific mechanism: how massive value in QK drives semantic understanding. We appreciate the opportunity to clarify this point.
>
> [1] Allen-Zhu and Li. Physics of Language Models: Part 1, Learning Hierarchical Language Structures.
>
> [2] Liu Z, Yuan J, et al. KIVI: A Tuning-Free Asymmetric 2bit Quantization for KV Cache. In ICML.
>
> [3] Jin H, Han X, et al. LLM Maybe LongLM: SelfExtend LLM context window without tuning. In ICML.
>
> [4] Barbero F, Vitvitskyi A, et al. Round and Round We Go! What makes Rotary Positional Encodings useful? In ICLR.

---

### Official Review · Reviewer_aSeD · 2025-03-14

**Overall Recommendation:** 3

**Summary:**

This paper investigates the emergence of massive values in self-attention within LLMs and their impact on contextual knowledge understanding. Through extensive experiment, the authors demonstrate that these massive values play a crucial role in interpreting contextually provided knowledge rather than retrieving parametric knowledge stored in the model’s parameters. They further show that these massive values arise due to RoPE and persist across multiple layers. Additionally, the paper explores how quantization strategies that ignore these massive values can significantly degrade performance on tasks requiring contextual understanding.

**Claims And Evidence:**

The claims are generally well-supported by experiments, particularly through controlled ablations that show the causal impact of massive values on contextual knowledge understanding. However, further analysis of how massive values influence attention patterns and token interactions would strengthen the argument.

A notable limitation is the evidence supporting the claim that RoPE directly causes the emergence of massive values. While the authors correlate the presence of massive values with models that use RoPE (Table 3), this evidence is not entirely conclusive, as different models undergo distinct training procedures that may introduce confounding factors. A stronger validation could involve controlled experiments comparing architectures with and without RoPE under identical training conditions.

**Essential References Not Discussed:**

The paper sufficiently discusses relevant prior work.

**Experimental Designs Or Analyses:**

The disruption experiments convincingly show the causal impact of massive values on contextual knowledge understanding. However, additional visualization of attention patterns before and after disruption would make the results more interpretable. Also, the analysis could benefit from further exploration of which specific attention heads or layers contribute most to contextual understanding via these massive values.

**Methods And Evaluation Criteria:**

The authors conduct a thorough analysis using:

+ Diverse LLMs (e.g., LLaMA, Gemma, Qwen) to ensure generalizability.
+ Controlled ablation studies (disrupting massive values selectively) to isolate their impact.
+ A range of contextual and parametric knowledge tasks to validate the importance of massive values.
+ Comparative analysis of quantization strategies, offering practical insights into model optimization.

One limitation is that the paper does not explicitly investigate how massive values shape attention distributions and their direct influence on token importance within the context. Analyzing attention heatmaps or token importance shifts due to massive values would enhance the interpretability of the findings.

**Other Comments Or Suggestions:**

See above.

**Other Strengths And Weaknesses:**

Strengths
+ Novel and interesting discovery about the role of massive values in Q/K.
+ Comprehensive experiments demonstrating clear causal effects.
+ Practical implications for LLM quantization and optimization.

Weaknesses
+ Lacks a deep analysis of attention pattern changes due to massive values.
+ The theoretical justification could be made more formal.

**Questions For Authors:**

See above.

**Relation To Broader Scientific Literature:**

This paper connects well with prior work on:

+ Quantization-aware optimization (e.g., AWQ, SmoothQuant), showing how activation outliers affect model performance.
+ LLM representation analysis, particularly studies on RoPE and attention patterns.
+ Contextual vs. parametric knowledge distinction in LLMs, contributing new empirical evidence to this ongoing discussion.

A useful extension would be comparing findings to recent studies on attention head importance and pruning to see if massive values correlate with known influential heads.

**Theoretical Claims:**

The paper does not provide any formal theoretical claims regarding the emergence or impact of massive values.

---

> ### Author Rebuttal · Authors · 2025-03-30
>
> > W1: Lacks a deep analysis of attention pattern changes due to massive values.
>
> We thank the reviewer for pointing this out. We agree that a deeper analysis of how massive values affect attention patterns would strengthen the paper, and we will include such an analysis in the revision. Specifically, we have conducted a new experiment that compares attention scores under two settings:
>
> 1. **With massive values intact**, and
> 2. **With massive values disrupted**.
>
>
> We computed the average attention scores across all tokens in one sentence in both settings and observed that:
>
> - When massive values are **not disrupted**, the LLM produces **higher attention scores** per token, reflecting concentrated and confident attention.
> - When massive values are **destroyed**, attention scores become **much lower and diffuse** per token, indicating weaker attention focus.
>
> To ensure robustness, we performed this experiment on three datasets (GSM8K, AQUA, and IMDB) using LLaMA3–8B. For each dataset, we sampled 100 examples and measured the token-level average attention score difference between the two settings.
> The new experiment shown below as the table provides concrete evidence that massive values significantly shape the model’s attention behavior.
>
> | Dataset | Model     | Intact Attention Score (Average) | Destroyed Attention Score (Average)  | Difference (Intact − Destroyed) | Notes                        |
> |---------|-----------|--------------|-----------------|----------------------------------|------------------------------|
> | GSM8K   | LLaMA3–8B | -0.1028      | -0.5039         | 0.4011                           | Higher when massive intact   |
> | AQUA    | LLaMA3–8B | -0.0633      | -0.3748         | 0.3115                           | Higher when massive intact   |
> | IMDB    | LLaMA3–8B | -0.1837      | -0.4919         | 0.3082                           | Higher when massive intact   |
>
>
> As shown, **the average attention score significantly decreases when massive values are disrupted across all datasets**. This highlights their functional role in massive value in LLM. We will include the analysis and visualizations in the updated version.
>
>
>
>
>
>
>
> > W2: The theoretical justification could be made more formal.
>
> We appreciate the reviewer's feedback regarding the formality of the theoretical justification and agree that more rigorous theoretical proof would strengthen the paper. However, indeed, the theoretical understanding of large language models, particularly regarding internal mechanisms like attention concentration and the emergence of massive value/ activation, is still in its early stages [1, 2, 3]. Our primary contribution lies in the empirical demonstration of how banded massive values emerge in the QK embedding vectors and how they functionally impact attention behavior. We show that these values consistently align with semantically important tokens and drive attention concentration. Furthermore, through carefully controlled interventions, we isolate the role of RoPE in forming these banded patterns.  As noted in several related works, many empirical studies in this area also primarily rely on observations and interventions rather than formal proofs or theoretical frameworks due to modern transformer models' inherent complexity and scale [1, 2, 3, 4].
>
>
> Therefore, we thank the reviewer for this valuable suggestion and leave the theoretical proof as our future work.
>
> [1] Sun M, Chen X, Kolter J Z, et al. Massive Activations in Large Language Models. In COLM 2024.
>
> [2] Barbero F, et al. Round and Round We Go! What makes Rotary Positional Encodings useful? In ICLR 2025.
>
> [3] Yu M, Wang D, Shan Q, et al. The super weight in large language models. arXiv preprint 2024.
>
> [4] Zhang S, Papyan V. Oats: Outlier-aware pruning through sparse and low rank decomposition. arXiv preprint 2024

---

### Official Review · Reviewer_3Gju · 2025-03-16

**Overall Recommendation:** 3

**Summary:**

This paper investigates the massive value elements in output vectors of internal layers of Transformer from the viewpoint of their influences on task-solving performance and the reason for them. The empirical analysis of various large language models (LLMs) reveals that the massive values influence tasks requiring context understanding rather than tasks requiring stored knowledge in pre-training. Furthermore, by conducting an empirical ablation study across different LLMs, the authors obtain a result indicating the relationship between Rotary Positional Encoding (RoPE) and massive values.

## update after rebuttal

The authors shared some details about hyperparameters through the discussion, which must improve the reproducibility of this paper. However, the fundamental issue of the lack of theoretical explanation remains unresolved. Thus, I keep the current updated score.

**Claims And Evidence:**

The claim for the existence of the massive values in LLMs and RoPE as their reason is actually shown in the empirical results based on various LLMs with various datasets. However, the authors seem to arbitrarily choose $\lambda$, which decides the threshold of the massive value, and it may raise some questions about the observed results.

**Essential References Not Discussed:**

Related works are appropriately cited.

**Experimental Designs Or Analyses:**

I think it's appropriate except for some hyper-parameters. I'll decide its validity through the discussion phase.

**Methods And Evaluation Criteria:**

It makes sense since the authors used well-known benchmark datasets and their standard metrics.

**Other Comments Or Suggestions:**

The last paragraph in subsection 3.5: (Yu et al.) -> (Yu et al. 2023)

**Other Strengths And Weaknesses:**

Strengths:
- This work reveals the massive value is more influential on tasks requiring context understanding rather than tasks require stored knowledge in pre-training.
- The empirical analysis also reveals the relationship between RoPE and the massive value.
- Experimental results are based on various models with various datasets.

Weaknesses:
- How the massive value occurs is still uncertain.
- In contrast to the empirical analysis, this work lacks theoretical aspects.
- Some settings, like the tuning of $\lambda$ and how to define the non-massive value, are not well described.

**Questions For Authors:**

- How did you set $\lambda$? If you tuned it on validation datasets, please teach the used model for the decision.
- What is the definition of the non-massive value in Table 1?
- How did you choose the value (dimension) for disrupting the non-massive value in Table 1?

**Relation To Broader Scientific Literature:**

The massive value becomes an obstacle in quantization and weight pruning. Thus, this research influences various NLP tasks on LLMs when these require efficiency.

**Theoretical Claims:**

Almost all claims are justified by empirical experiments. Thus, there is no theoretical proof.

---

> ### Author Rebuttal · Authors · 2025-03-29
>
> > S1: subsection 3.5: (Yu et al.) -> (Yu et al. 2023)
>
> Thank you for your suggestion. We will correct the citation in subsection 3.5 from (Yu et al.) to (Yu et al. 2023) to ensure proper citation formatting.
>
> > Q1: How the massive value occurs is still uncertain.
>
> Thank you for raising this critical point. The global phenomenon of massive values in LLMs is still under active investigation, our contribution does not aim to resolve this broader uncertainty fully. Instead, our work focuses on a specific and consistent pattern we identify within the QK Embedding Vector: the presence of banded, stripe-like massive values that arise due to the structure of RoPE. We fully agree that the origin of massive values in transformer attention mechanisms remains an open question. Prior work has observed their existence in large language models but did not provide a complete explanation [1, 2, 3]. It did not explain some questions, like how massive values appear in the training process.
>
> > Q2: In contrast to the empirical analysis, this work lacks theoretical aspects.
>
> We acknowledge the concern regarding the lack of theoretical analysis. Indeed, the theoretical understanding of LLM, particularly regarding internal mechanisms like attention concentration and the emergence of massive values in QK, is still in its early stages [1, 3].  As noted in several related works, many empirical studies in this area also primarily rely on observations and interventions rather than formal proofs or theoretical frameworks due to the inherent complexity and scale of modern transformer-based LLM [3]. Therefore, we thank the reviewer for this valuable suggestion and will add the theoretical proof for our future work.
>
>
> > Q3: How did you set λ? If you tuned it on validation datasets, please teach the used model for the decision.
>
> Thank you for your insightful question. We would like to clarify that the threshold parameter λ = 5 was not tuned on any validation set. Instead, this value was selected empirically based on statistical observations across all models and datasets used in our experiments. Concretely, we analyzed the query and key embeddings (x-axis is head, y-axis is dim) across all layers of three LLMs on three datasets (GSM8K, AQUA, and IMDB). **We consistently observed that the massive values in each head were typically at least 5× larger than the mean value of its head. This motivated our choice of λ = 5 as a conservative and robust threshold to capture the most prominent outliers.** We also confirmed that moderate variations around this threshold (e.g., λ ∈ [5, 6]) did not qualitatively affect our findings. We will include a clarification in the final version and optionally add this robustness observation to Appendix F. The following table summarizes the models, layers, and datasets we used in the statistical analysis.
> | Model       | Layer   | GSM8K | AQUA | IMDB |
> |-------------|---------|-------|------|------|
> | Gemma2–9B   | Layer 1 | √     | √    | √    |
> |             | Layer 15 | √     | √    | √    |
> |             | Layer 27| √     | √    | √    |
> | Llama3–8B   | Layer 1 | √     | √    | √    |
> |             | Layer 15 | √     | √    | √    |
> |             | Layer 27| √     | √    | √    |
> | Qwen2.5–7B  | Layer 1 | √     | √    | √    |
> |             | Layer 15 | √     | √    | √    |
> |             | Layer 27| √     | √    | √    |
>
>
> > Q4: What is the definition of the non-massive value in Table 1?
>
> Thank you for your insightful question. In Table 1, "non-massive value" refers to the value **selected from the non-extreme regions of the Q/K embedding vector**, i.e., value in dimensions outside specific dimensions where concentrated values occur. In the next version, we will clarify this definition more explicitly to avoid ambiguity.
>
>
> > Q5: How did you choose the value (dimension) for disrupting the non-massive value in Table 1?
>
> This point is closely related to Q4, and we appreciate the opportunity to clarify it further. To select the non-massive value, we first analyzed the Q/K embedding vector's shape (with heads along the x-axis and dimensions along the y-axis). For each head index in the x-axis, we excluded the region we previously identified as containing massive values. Then, **we randomly selected one dimension outside the region where concentrated massive values occur to choose non-massive values in this head**. This process is repeated for each head index along the x-axis. We then applied the same disruption method: setting the selected dimension to the mean value. We will describe this procedure more clearly and explicitly in the next revision. Thank you for pointing this out.
>
>
>
>
>
>
>
>
>
>
> [1] Sun M, et al. Massive Activations in Large Language Models. In COLM.
>
> [2] Wang T, Liu Y, et al. M2PT: Multimodal Prompt Tuning for Zero-shot Instruction Learning. In EMNLP.
>
> [3] Barbero F, et al. Round and Round We Go! What makes Rotary Positional Encodings useful? In ICLR.

---

### Official Review · Reviewer_XgBB · 2025-03-17

**Overall Recommendation:** 3

**Summary:**

The authors study the case of Massive values K,Q values which appear in LLMs.
The authors conclude that the primary reason for this ROPE embeddings.
Another interesting finding the authors proposed was that the massive K,Q values have different impact in different types of task.

**Claims And Evidence:**

The authors make following key claims -

Massive values are concentrated in specific regions of Q and K exclusively

Massive values in Q and K are critical for understanding contextual knowledge over parametric knowledge.

Quantization techniques targeting massive values preserve contextual knowledge better

Concentration of massive values is caused by RoPE and it appears since very early layers in LLMs. -> How does not having RoPE affect the accuracy.

**Essential References Not Discussed:**

There are quite a few works which are inherently dependent on similar insights. For example - Efficient streaming language models with attention sinks, look at similar insights that different position encoding locations.

**Experimental Designs Or Analyses:**

Yes. Checked the study design for studying different dataset performance.

**Methods And Evaluation Criteria:**

The authors have created a reasonable study setup to verify the claims

**Other Comments Or Suggestions:**

None

**Other Strengths And Weaknesses:**

- I think the study is quite interesting, however the impact is quite unclear. For example- Existing optimization use similar insights. However, this does not clearly answer the question of why RoPE embeddings are causing this. Secondly it does not compare how not having these large values in case of Jamba/GPT-2 affects performance.

- It will be interesting if authors can perform a study with different types of embeddings like AliBi

**Questions For Authors:**

See weakness and strength section.

**Relation To Broader Scientific Literature:**

Several prior works like Cachegen, StreamingLLMs have similar insights. My primary concern is that some of these findings are not new.

**Theoretical Claims:**

There are no theoretical claims in this paper.

---

> ### Author Rebuttal · Authors · 2025-03-30
>
> > W1: Several prior works like Cachegen StreamingLLMs have similar insights. My primary concern is that some of these findings are not new. I think the study is quite interesting, however the impact is quite unclear. For example- Existing optimization uses similar insights.
>
> Thank you for pointing out these related works [1, 2]. While these works do explore caching mechanisms in LLM inference, we must emphasize that our research differs fundamentally from these prior works in focus: CacheGen and StreamingLLMs primarily investigate the positional dynamics of tokens in attention mechanisms, analyzing how massive attention values at specific tokens impact model performance. In contrast, our research specifically examines the **massiva values in the Q and K embedding vectors along the embedding depth of tokens**, which analyzes massive values in a different subspace to provide complementary insights to existing approaches.
>
> > W2: This does not clearly answer the question of why RoPE embeddings are causing this. Secondly it does not compare how not having these large values in case of Jamba/GPT-2 affects performance.
>
> We would like to clarify that we **concentrate on identifying and explaining a specific pattern of massive values that consistently appears in models using RoPE** while we **do not explain the origin of massive values in transformers in general**. Prior works [5, 6] have observed the existence of massive values in large language models but did not provide a complete explanation of how these massive values emerge during the training process. Our contribution does not aim to resolve this broader problem fully.
> From the results we observed in Jamba, GPT-2 and OPT shown in Table 3 of our paper, concentrated massive values could only be observed in models employing RoPE. To further validate this, we conduct experiments across three models with similar architectures but different positional embedding methods: GPT-2, GPT-2-neo, and GPT-2-neox to just focus on the impact of positional embedding differences.
>
> | Model  | Position Embedding | Concentrated Massive Value |
> |--------|------------------|------------------|
> | GPT-2   | Absolute | No  |
> | GPT-2-Neo   | Absolute | No |
> | GPT-2-NeoX   | RoPE | Yes |
>
>
> The results show that concentrated massive values can only be found in  GPT-2-NeoX employing RoPE, which is consistent with what we have observed.  In addition, we conduct experiments of disrupting all the massive values in OPT and Jamba on GSM8k and Cities, respectively (we do not consider GPT-2 as it does have the capability to fulfill any of the tasks. )
>
> | Model  | Massive Value Disrupted | GSM8k (%) | Cities (%) |
> |--------|------------------|------------------|------------------|
> |  OPT-2.7B   | No | 5.0  | 76.0  |
> | OPT-2.7B  | Yes  | 2.0  | 60.1  |
> | Jamba-1.5-mini   | No | 72.8  | 96.5  |
> | Jamba-1.5-mini   | Yes | 50.2  | 75.2  |
>
>
> The results show obvious **performance degradation in both in-context retrieval and parametric retrieval when disrupting massive values in models without using RoPE**. This implies that unlike RoPE where positional and semantic information could be separated into high and low dimensions of Q and K embedding vectors, massive values in models without RoPE appear to blend with positional and semantic information.
>
> > W3: It will be interesting if authors can perform a study with different types of embeddings like AliBi.
>
> Thank you for mentioning this point. We conducted this exact comparison between RoPE and ALiBi using Baichuan-2-7B (using RoPE) and Baichuan-2-13B (using ALiBi) [3]. The results can be seen from the following table.
>
> | Model  | Position Embedding | Concentrated Massive Value |
> |--------|------------------|------------------|
> | Baichuan-2-7B   | RoPE | Yes  |
> | Baichun-2-13B   | AliBi | No |
>
> The results revealed that **concentrated massive values can be observed in Baichuan-7B while absent in Baichuan-13B. This difference can be attributed to the fundamental mechanisms of these positional embeddings**. Unlike RoPE, ALiBi applies a simple linear bias based on relative token distances, which maintains a more uniform attention distribution that inhibits the emergence of concentrated massive values. Complete experimental results will be included in the updated version of our paper.
>
> [1] Liu et al. Cachegen: Kv cache compression and streaming for fast large language model serving. 2024 Sigcomm
>
> [2] Xiao et al. Efficient streaming language models with attention sinks. 2023
>
> [3] Yang et al. Baichuan 2: Open Large-scale Language Models. 2023
>
> [4] Barbero F, Vitvitskyi A, Perivolaropoulos C, et al. Round and Round We Go! What makes Rotary Positional Encodings useful? In ICLR 2025.
>
> [5] Jianlin Su, Murtadha Ahmed, Yu Lu, et al. Roformer: Enhanced transformer with rotary position embedding. Neurocomputing, 568:127063, 2024.
>
> [6] Sun M, Chen X, Kolter J Z, et al. Massive Activations in Large Language Models. In COLM 2024.

---

### Decision · Program_Chairs · 2025-05-01

**Decision:**

Accept (poster)

**Comment:**

This work presents an interesting observation regarding large values in the attention mechanism: specifically, the query and key components exhibit large magnitudes, while the value component does not. The authors empirically attribute this behavior to the use of ROPE embeddings. Experimental results suggest that this dynamic plays an important role in answering contextual questions, as opposed to retrieving parametric knowledge. However, the discussion of related literature is incomplete in the initial submission, though the authors have promised to address this in the camera-ready version. Another notable weakness is the lack of in-depth analysis explaining why this phenomenon occurs.